# Towards flexible perception with visual memory

## Abstract

Training a neural network is a monolithic endeavor, akin to carving knowledge into stone: once the process is completed, editing the knowledge in a network is nearly impossible, since all information is distributed across the network's weights. We here explore a simple, compelling alternative by marrying the representational power of deep neural networks with the flexibility of a database. Decomposing the task of image classification into image similarity (from a pretrained embedding) and search (via fast nearest neighbor retrieval from a knowledge database), we build a simple and flexible visual memory that has the following key capabilities: (1.) The ability to flexibly add data across scales: from individual samples all the way to entire classes and billion-scale data; (2.) The ability to remove data through unlearning and memory pruning; (3.) An interpretable decision-mechanism on which we can intervene to control its behavior. Taken together, these capabilities comprehensively demonstrate the benefits of an explicit visual memory. We hope that it might contribute to a conversation on how knowledge should be represented in deep vision models—beyond carving it in "stone" weights.

## 1 Introduction

In the pretty diagrams on "Intro to Machine Learning" slides, an ideal ML workflow looks like this: Data collection, preprocessing, choosing a model, training, evaluation, deployment. Happy ending—the model is deployed, the users love it, and one can finally go on that well-deserved vacation and catch up on the latest AGI memes.

Until, of course, the enemy of any ideal world sets in: reality. The real world constantly keeps changing, and so do data requirements. New data and datasets become available, and existing ones become deprecated for a variety of reasons, including concerns around fairness, biases or unsafe content. Knowledge changes, and concepts drift (Tsymbal, 2004; Lu et al., 2018): Phones and cars look different today than they did a few years ago, and different from how they will look in the future. When it comes to data, the only constant is change (Cao & Yang, 2015; Bourtoule et al., 2021; Nguyen et al., 2022; Zhang et al., 2023). Consequently, from a modeling perspective, in order to keep up with this change one would ideally want to constantly re-train or fine-tune models, which is of course not feasible. In short, as anyone who has ever deployed a model has experienced firsthand, one is constantly battling the symptoms of a single underlying cause: the fact that deep learning models have a static knowledge representation entangled in millions or billions of model parameters. We, among many others working on memory (e.g. Weston et al., 2014; Chen et al., 2018; Wu et al., 2021; Iscen et al., 2022; Nakata et al., 2022; Iscen et al., 2023; Prabhu et al., 2023; Gui et al., 2024; Shao et al., 2024; Silva et al., 2024), believe that this is not a great way to represent visual knowledge for deep learning. Instead, we argue that we should build models that cleanly separate representation (*how* things are represented, e.g. through feature embeddings) from visual memory (*what* is known). In short, deep learning models need a flexible visual memory: a way to explicitly utilize and edit knowledge.

In this work, we build a simple visual memory for classification and show that it has seven desirable capabilities, including the ability to flexibly add data across scales (from individual samples to classes and even billion-scale data), the ability to remove data from our model's classification process through machine unlearning and memory pruning, and a simple, interpretable decision-

mechanism on which we can intervene to control its behavior. Our main goal is to provide a compelling idea of how beneficial a flexible visual memory for deep learning can be from a variety of perspectives and capabilities. From a technical standpoint, we aim for simplicity: retrieving $k$ nearest neighbors (in an embedding feature space) along with their labels to classify a query image. This approach allows us to investigate where a simple visual memory mechanism helps, where its limitations may be, and where there might be opportunities for improvement through a more complex system. We hope that by demonstrating clear benefits from a simple visual memory, this article might contribute to a conversation on how knowledge ought to be represented in deep vision models.

Here are some highlights of this article:

1. Improved aggregation of retrieved samples: we propose using *RankVoting*, a power-law weighting that surpasses previous SOTA (SoftmaxVoting) for a deep learning based memory.

2. Re-ranking samples using a vision-language model achieves 88.5% top-1 ImageNet validation accuracy, improving over both DinoV2 ViT-L14 kNN and linear probing.

3. Flexible perception: the visual memory achieves perfect unlearning, scales to billion-scale data without additional training, and enables controlling sample influence via *memory pruning*.

We argue that the way current deep learning models represent knowledge (static knowledge representation, hard to update, hard to unlearn, hard to understand how a decision is made) is problematic. As an alternative, we built a working proof-of-concept: By building on the long history of nearest neighbor methods, and "marrying" them with a powerful deep learning representation (such as SSL features from DinoV2) and a billion-scale visual memory.

**Related work.** The concept of a visual memory has a long history in ML, neuroscience and psychology. In psychology, *exemplar theory* posits that humans recognize objects by comparing them to existing examples in visual memory (Medin & Schaffer, 1978; Nosofsky, 1986; Dopkins & Gleason, 1997; Jäkel et al., 2008; Nosofsky, 2011), like the ALCOVE model (Kruschke, 2020). In ML, prior to deep learning, *instance-based learning* (also known as memory-based learning) was a popular alternative to *model-based learning* (Aha et al., 1991; Quinlan, 1993). For instance, Turk & Pentland (1991) used nearest neighbor methods to classify faces, and Sivic & Zisserman (2003) build a visual memory inspired by text retrieval for object retrieval from videos. In recent years, hybrid approaches have started to combine the benefits of both approaches. Deep neural network variants (model-based since they learn generalized abstractions of data) of $k$-nearest neighbor algorithms (instance-based since they compare new data to existing exemplars in memory) have been proposed with various motivations, including few-shot learning (Wang et al., 2019b; Yang et al., 2020; Bari et al., 2021), improving adversarial robustness (Sitawarin & Wagner, 2019; Papernot & McDaniel, 2018; Rajani et al., 2020), medical image classification (Zhuang et al., 2020), confidence calibration (Papernot & McDaniel, 2018), interpretability (Papernot & McDaniel, 2018; Wallace et al., 2018; Lee et al., 2020; Rajani et al., 2020), image denoising (Plötz & Roth, 2018), retrieval-augmented learning (Khandelwal et al., 2019; Drozdov et al., 2022), anomaly and out-of-distribution detection (Bergman et al., 2020; Sun et al., 2022). Recently, Nakata et al. (2022) tested a kNN-based visual memory up to ImageNet-scale (1.28M images), and Khandelwal et al. (2019); Wu et al. (2021) applied kNN-based approaches to neural language models.

## 2 BUILDING A RETRIEVAL-BASED VISUAL MEMORY FOR CLASSIFICATION

Given a dataset $\mathcal{D}_{\text{test}} := \{(\tilde{x}_1, y_1), \cdots, \tilde{x}_n, y_n\}$, we want to classify each image $\tilde{x}_i \in \mathcal{D}_{\text{test}}$. Our classification approach consists of two steps: (i) building a visual memory, and (ii) fast nearest neighbor based inference using the visual memory.

### 2.1 BUILDING A VISUAL MEMORY

Our visual memory retrieves (image, label) pairs from an image dataset when a query is made by directly retrieving those images that are considered similar to a test image according to a model. The model is a fixed pre-trained image encoder, meaning that no training takes place when adding information to visual memory. No copies of the dataset are stored in the visual memory. Instead, feature maps are extracted from the model based on a set of images related to the downstream classification task at hand, such as a standard training set. For our experiments, our visual memory

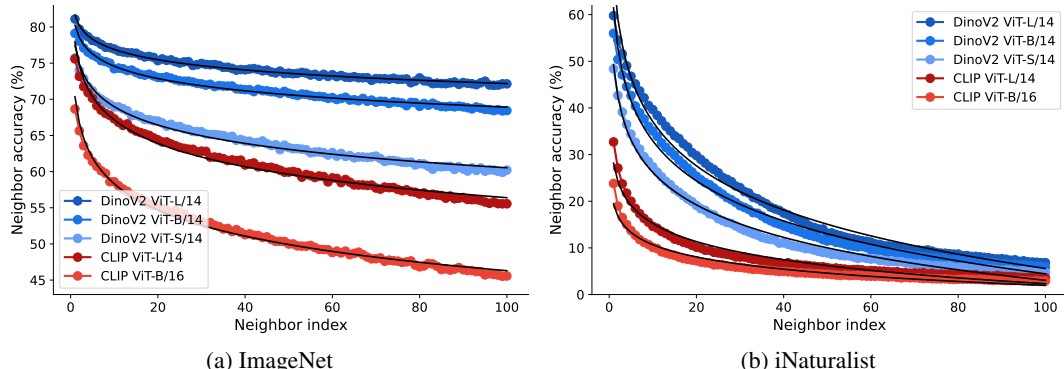

(a) ImageNet                    (b) iNaturalist

Figure 1: **Reliability of retrieved memory samples.** This plot visualizes the ImageNet **(left)** and iNaturalist **(right)** top-1 validation accuracy of a single retrieved neighbor depending on the index of the neighbor (index 0: nearest neighbor). In both datasets and across models, the decrease in accuracy with increasing neighbor index follows smooth trajectories and can be approximated by a two-parameter logarithmic fit (black lines).

comprises of features extracted from a dataset like the ImageNet-1K (Russakovsky et al., 2015) training set using different encoders like DinoV2 (Oquab et al., 2023) and CLIP (Radford et al., 2021). Thus, given a pretrained image encoder, $\Phi$, and a dataset of (image, label) pairs $\mathcal{D}_{\text{train}} := (\boldsymbol{x}_1, y_1), (\boldsymbol{x}_2, y_2), \cdots, (\boldsymbol{x}_N, y_N)$, we obtain features $\boldsymbol{z}_i := \Phi(\boldsymbol{x}_i), \forall \boldsymbol{x}_i \in \mathcal{D}_{\text{train}}$. Subsequently, the feature maps and corresponding label pairs are put in a database thereby creating VisualMemory $:= \{(\boldsymbol{z}_1, y_1), (\boldsymbol{z}_2, y_2), \cdots, (\boldsymbol{z}_N, y_N)\}$ for classification. For both DinoV2 and CLIP, we use the last image embedding layer as a feature space.

## 2.2 RETRIEVAL-BASED CLASSIFICATION USING VISUAL MEMORY

Given a query image $\tilde{\boldsymbol{x}} \in \mathcal{D}_{\text{test}}$, we extract its feature map, $\tilde{\boldsymbol{z}} = \Phi(\tilde{\boldsymbol{x}})$. We then query VisualMemory to extract $k$ feature vectors, Neighbors$(\tilde{\boldsymbol{x}}) := \{(\boldsymbol{z}_{[1]}, y_{[1]}), (\boldsymbol{z}_{[2]}, y_{[2]}((, \cdots, (\boldsymbol{z}_{[k]}, y_{[k]})\}$, that are closest to the query features $\tilde{\boldsymbol{z}}$ using the cosine distance, which is the default retrieval similarity measure for SSL models like DinoV2. Neighbors$(\tilde{\boldsymbol{x}})$, are ordered by distance i.e.

$$\text{dist}(\tilde{\boldsymbol{z}}, \boldsymbol{z}_{[i]}) \leq \text{dist}(\tilde{\boldsymbol{z}}, \boldsymbol{z}_{[j]}), \ \forall i \leq j.$$

We then assign a weight, $w_i$, to each neighbour $(\boldsymbol{z}_{[i]}, y_{[i]})$ and aggregate the scores for each neighbour with the same label. Finally, we assign that label to the query image with the highest aggregate score. We implemented retrieval based classification using one of the following two approaches:

**1. Fast inference using matrix multiplication on GPUs/TPUs:** For smaller datasets like ImageNet, we saved VisualMemory as a matrix of size num_images $\times$ num_dims. During inference, for an encoded query image of size $1 \times$ num_dims, we computed the dot product of this encoded image with every entry in VisualMemory getting a matrix of size num_images $\times 1$. We then computed the $k$ nearest neighbors using the $\arg\max$ operation.

**2. Fast and scalable nearest neighbor search:** We used ScaNN (Guo et al., 2020) for accelerating nearest neighbor search at scale. Specifically, we saved the VisualMemory as a database and used ScaNN for fast lookup of nearest neighbors during inference. This method scales easily to billion-scale memory (cf. Section 3.3). Appendix J details latency and storage; storing features requires only about 1–3% of the space of storing the dataset itself and even with a 1B memory.

We mentioned earlier that we retrieve a set of neighbors, Neighbors$(\tilde{\boldsymbol{x}})$ and aggregate information across them to make a classification decision. In order to understand how reliable (i.e., accurate) retrieved memory samples are from the first to the 100th neighbor, we systematically analyze neighbor reliability in Figure 1. As expected, reliability decreases as the neighbor index $k$ increases, but even at large $k$ the neighbors contain above-chance information about the ground truth class. This suggests that aggregating information across different neighbors may be beneficial to decision-making, leading to the question: *What is the best aggregation strategy?* We empirically study this by testing different weighting strategies for aggregation:

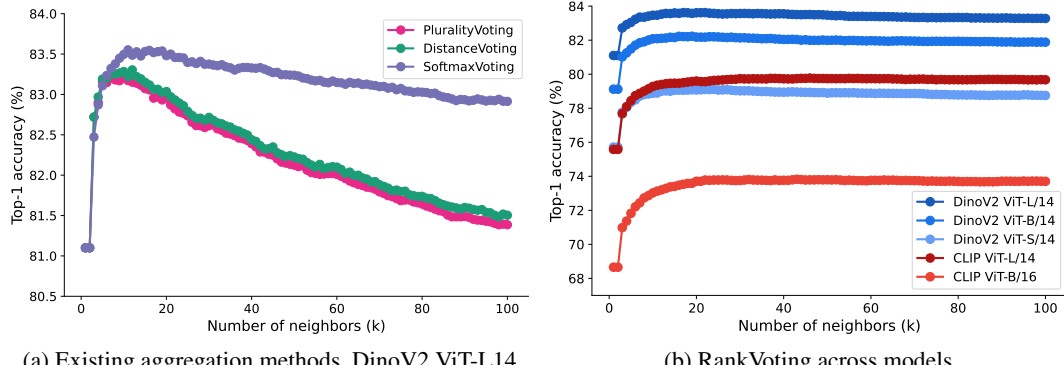

(a) Existing aggregation methods, DinoV2 ViT-L14  (b) RankVoting across models

Figure 2: **Aggregating information across retrieved memory samples. (left)** Existing aggregation methods are overconfident in distant neighbors, resulting in the paradox of decaying ImageNet-1K accuracy with more information. The same pattern is also seen for other models and datasets in the Appendix (Figures 7 and 8). **(right)** This is not the case for RankVoting, a simple power-function based method which reaches higher and stable performance across models and choices of $k$.

**Plurality voting:** Each neighbour in Neighbors($\tilde{x}$) is assigned an equal weight of 1.0. This is the classic, most simple voting method and used for instance by Nakata et al. (2022).

**Distance voting:** Each neighbour in Neighbors($\tilde{x}$) is assigned a weight based on its Cosine distance to the query image $\tilde{x}$ i.e. $w_i = \exp\big(-\mathsf{dist}(\tilde{z}, z_{[i]})\big)$. This approach has been used by Khandelwal et al. (2019) for nearest neighbor language models.

**Softmax voting:** Each neighbour is assigned a weight based on the softmax function i.e. $w_i = \mathsf{softmax}\big(\mathsf{dist}(\tilde{z}, z_{[i]}), \tau\big)$ where $\tau$ is the temperature. This voting method is considered state-of-the-art; for example nearest neighbor accuracies of self-supervised models are reported using this method. A temperature of $\tau = 0.07$ frequently appears in literature (Wu et al., 2018; Caron et al., 2021; Oquab et al., 2023) and is reported as a parameter "which we do not tune" in the Dino paper (Caron et al., 2021, p. 18). We observe that performance is sensitive to this parameter; other temperatures perform worse. We therefore follow the literature in using $\tau = 0.07$.

**Rank voting:** We propose using a simple aggregation approach wherein each neighbour is assigned a power-function weight based on its rank in the ordered set Neighbors($\tilde{x}$) i.e. $w_i = 1/(\alpha + \mathsf{rank}_i)$ where $\mathsf{rank}_i$ is $i$ and $\alpha$ is an offset to avoid division by zero that is set to 2.0. This is similar, though not identical to, Gou et al. (2011) who used power-law weighting in a different context.

In Figure 2a, we compare the top-1 ImageNet validation accuracy of different ranking methods as a function of number of neighbours, with the ImageNet-1K training set as the visual memory using the DinoV2/ViT-L14 model as the featurizer. Paradoxically, existing aggregation methods like plurality voting, distance-based voting, and softmax voting show *decaying* performance as the provided information (number of nearest neighbors) *increases*. This suggests that the methods are overconfident in distant neighbors, assigning them too much weight. Our simple, parameter-free rank based voting method, however, leads to an increase in performance with more neighbors until a certain $k$ after which the performance plateaus, which is the ideal scenario (Figure 2b). Furthermore, rank-based voting also outperforms baselines in absolute terms; quantitative comparisons can be found in the Appendix (Tables 4 to 8) where we also study the influence of hyperparameters (Figure 9). This indicates that a simple, power-function based method can reliably integrate information across retrieved memory samples.

**Gemini re-ranking.** Our results above demonstrate that different aggregation strategies have a large impact on downstream performance. How far can we push the upper limit on aggregating information from different neighbors? We perform a controlled experiment using the Gemini 1.5 Flash model (Reid et al., 2024) to test this: We add the 50 nearest neighbors from DinoV2 ViT-L14 for a query image along with their labels into Gemini's context. We then query Gemini to predict the query image's label. This achieves 88.5% ImageNet validation accuracy, a substantial improvement over both DinoV2 ViT-L14 kNN (83.5%) and linear probing (86.3%) performance. Interestingly,

Gemini's performance is mainly driven by the neighbor information through in-context learning since it only achieves 69.6% accuracy without neighbors (when just the query image is provided to the model). The performance improvement highlights the potential of using vision-language models as a visual memory re-ranker. Given that our main goal is to explore a simple visual memory system, we mostly focus on non-Gemini ranking methods throughout our analysis.

## 3 CAPABILITIES OF A VISUAL MEMORY

Our primary goal is to motivate the concept of a machine *visual memory* from a variety of different perspectives. To this end, we investigate how such a memory can benefit the following capabilities: 3.1 Flexible lifelong learning: adding novel OOD classes; 3.2 Flexibly trading off compute and memory; 3.3 Flexibly adding billion-scale data without training; 3.4 Flexible removal of data: machine unlearning; 3.5 Flexible data selection: memory pruning; 3.6 Flexibly increasing dataset granularity; 3.7 Interpretable & attributable decision-making.

### 3.1 FLEXIBLE LIFELONG LEARNING: ADDING NOVEL OOD CLASSES (DATA AND LABELS)

Standard classifiers, whether trained end-to-end (supervised models) or with a linear classifier (self-supervised models), are not able to handle new information without re-training. For instance, adding new classes or changing labels in an existing model usually involves either re-training or fine-tuning parts of the model. A retrieval-based visual memory, in contrast, is able to process such information in a natural and flexible way, aligning with the requirements of lifelong learning (Parisi et al., 2019). We tested this by adding data for 64 new classes, along with their new labels, to the visual memory of a pre-trained DinoV2 ViT-L14 model (in addition to the ImageNet train set, which is in-distribution for the model). We took the new classes from the NINCO dataset (Bitterwolf et al., 2023), a dedicated OOD dataset that is designed to have no overlap with existing ImageNet labels and samples. This requires the model to transfer what it has learned to new, unseen concepts. The new task is therefore harder, as the model has to retrieve images from both in-distribution and OOD classes. The resulting visual memory has 1064 classes (1K from ImageNet and 64 from NINCO). Table 1 shows that with a visual memory it is possible to add new classes such that the in-distribution accuracy is maintained without catastrophic forgetting (the new classes only change ImageNet validation performance by 0.02–0.04% depending on the aggregation method), while at the same time reaching very high accuracy on the new OOD classes (approx. 87% top-1) without any training. Figure 12 in the appendix confirms that the samples are indeed OOD for the model, as demonstrated by larger distances to nearest neighbors. This highlights that a visual memory is capable of flexibly adding new information—an important capability since the world is not static. Furthermore, the memory is incredibly robust towards label corruption up to 60% random labels, as shown in Appendix D.

Table 1: **Flexible lifelong learning: adding novel OOD classes.** A visual memory of DinoV2 ViT-L14 with ImageNet-train (IN-train) as the memory database is able to handle a simple "insert into memory" operation for 64 out-of-distribution classes (data and labels) from the NINCO dataset (Bitterwolf et al., 2023), leading to high performance on the new classes without affecting top-1 ImageNet validation accuracy.

| memory → | IN-train | IN-train-and-NINCO | |
| query → | IN-val | IN-val | NINCO |
| --- | --- | --- | --- |
| no aggregation | 81.1 | 81.1 | 86.4 |
| PluralityVoting | 83.2 | 83.2 | 86.9 |
| DistanceVoting | 83.3 | 83.3 | 87.1 |
| SoftmaxVoting | 83.6 | 83.5 | 87.5 |
| RankVoting | 83.6 | 83.6 | 87.4 |

### 3.2 FLEXIBLY TRADING OFF COMPUTE AND MEMORY

Next, we turn our attention to studying the scaling behaviour of visual memory with increasing memory model size. We hypothesize that bigger models will be able to attain similar performance as

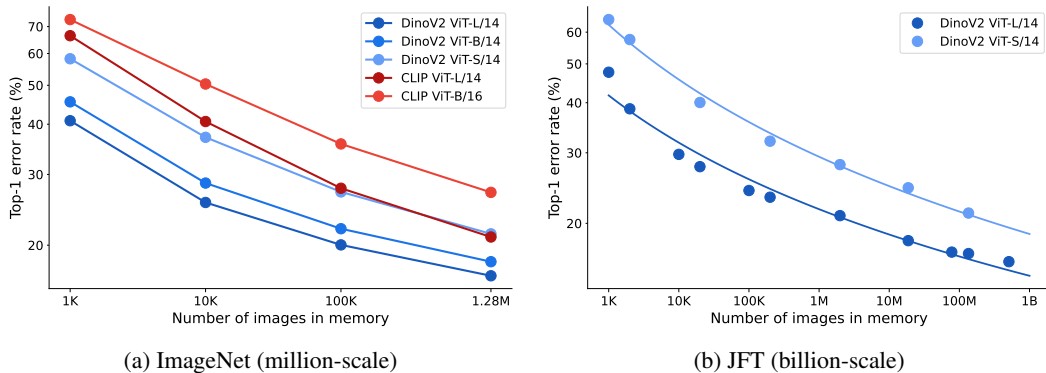

(a) ImageNet (million-scale)  (b) JFT (billion-scale)

Figure 3: **Memory scaling: flexibly trading off compute and memory.** ImageNet top-1 validation error decreases systematically as the memory size is increased (i.e., recognition accuracy increases with scale). **(left)** Million-scale memory consisting of ImageNet-train labels. **(right)** Billion-scale memory bank consisting of machine-generated pseudo labels on the JFT dataset (Zhai et al., 2022). Accuracy continues to decrease even with billion-scale data in memory. The roughly constant offset between models of different sizes suggests the possibility of a flexible trade-off: The same error rate can be achieved with a small model and large memory, or a large model and a small memory.

smaller models with lesser amount of visual memory. This is because, all else being equal, a bigger model should be a better featurizer that requires fewer examples in memory to represent different concepts. We empirically study the scaling behaviour of visual memory based retrieval systems in Figure 3a using models of different sizes like DinoV2 ViT models of sizes S/14 (21M params), B/14 (86M params), and L/14 (300M params), as well as CLIP ViT models of sizes B/16 and L/14. We plot the top-1 error rate as a function of number of images in visual memory. The plot demonstrates that for each model, the error rate consistently decreases as we increase the visual memory size. Notably, already with a single exemplar per class in memory, ImageNet validation performance is far beyond chance (41% top-1 error for DinoV2 ViT-L14). It also visualizes the possibility of a flexible trade-off between model size and memory size: e.g. for the different DinoV2 models, the S/14, B/14, and L/14 variant achieve similar performance at 1.28M, ∼150K, and ∼70K memory capacity respectively. In line with Nakata et al. (2022), this indicates that a smaller model with large memory can match the performance of a larger model with smaller memory.

### 3.3 FLEXIBLY ADDING BILLION-SCALE DATA WITHOUT TRAINING

**Billion-scale dataset with pseudo labels.** As demonstrated in Section 3.2, performance systematically improves with increased memory size across both small and large models. We here test how far this trend holds beyond relatively small-scale, well-curated settings like ImageNet-1K by scaling visual memory to the billion-scale unlabeled data regime. We obtain a large-scale dataset from the union of the ImageNet-1K train set and a subset of the JFT-3B dataset (Zhai et al., 2022). To this end, we treat JFT as an unlabeled dataset by ignoring its original labels and instead obtaining *pseudo labels* by running them through ViT-22B-224px (Dehghani et al., 2023), a highly performant classifier. We excluded images whose labels do not have a correspondence with the ImageNet labels.

**Scaling.** In Figure 3b, we show the downstream ImageNet validation performance of two DinoV2-ViTs as a function of memory size. The plot demonstrates that even in the billion-scale data regime, validation error decreases when increasing memory size without any training. The gain from more data is most prominent when having fewer samples in memory (e.g., going from 1 to 10 samples per class). In log-log space, a logarithmic function fits the empirical scaling trend well. In the literature, simple scaling trends such as the one we observe are powerful predictors of scaling behaviour for different model and dataset sizes (Hestness et al., 2017; Kaplan et al., 2020; Hoffmann et al., 2022).

**Out-of-distribution performance.** In order to understand whether the benefits of increased memory size transfer to out-of-distribution (OOD) data, we compared DinoV2 ViT-L14 once with ImageNet-train in memory and once with JFT pseudo-labels in memory. The models are evaluated on the ImageNet-A (Hendrycks et al., 2021), ImageNet-R (Hendrycks et al., 2020), ImageNet-Sketch

(Wang et al., 2019a), ImageNet-V2 (Shankar et al., 2020), and ImageNet-ReaL (Beyer et al., 2020) datasets. As an additional well-performing yet "inflexible" baseline, we report linear probing accuracies from the DinoV2 paper (Oquab et al., 2023). Table 2 shows that visual memory scaled with JFT data improves OOD performance across all datasets compared to an ImageNet-based visual memory. Gemini re-ranking again improves leads to performance gains. Overall, the finding that memory scale transfers to OOD improvements is important in the context of continual learning, where a flexible visual memory can easily incorporate newly available data that the model was not trained on and improve performance both in- and out-of-distribution.

Table 2: **OOD evaluation.** Out-of-distribution performance improves with larger visual memory size. Across all datasets, a visual memory with JFT memory outperforms ImageNet memory demonstrating advantages of scaling visual memory for OOD performance. Probe details: Appendix I.

| Model | Method | IN-A | IN-R | IN-Sketch | IN-V2 | IN-ReaL |
|---|---|---|---|---|---|---|
| DinoV2 ViT-L14 | linear probe | 71.3 | 74.4 | 59.3 | 78.0 | 89.5 |
| DinoV2 ViT-L14 | ImageNet memory | 58.8 | 62.8 | 61.5 | 75.6 | 87.1 |
| | + Gemini re-ranking | 68.4 | 72.3 | 72.5 | 81.7 | 89.9 |
| DinoV2 ViT-L14 | JFT memory | 61.1 | 73.7 | 68.0 | 77.6 | 88.2 |
| | + Gemini re-ranking | 69.6 | 81.4 | 75.0 | 82.3 | 90.5 |

## 3.4 FLEXIBLE REMOVAL OF DATA: MACHINE UNLEARNING

The world is not static. Thus, in addition to the need to flexibly add novel data, there is often a need to remove the influence of specific training data from a model's decision-making process after it has been trained (Cao & Yang, 2015; Bourtoule et al., 2021; Nguyen et al., 2022; Zhang et al., 2023). A range of intricate methods are being developed to remove or reduce the influence of certain training samples (Gupta et al., 2021; Sekhari et al., 2021; Ullah et al., 2021; Kurmanji et al., 2024; Sepahvand et al., 2024)—a challenging endeavour if knowledge is embedded in millions or billions of model weights. In contrast, for models with an explicit visual memory, machine unlearning becomes as simple as removing the dataset sample from the visual memory. For instance, after adding the NINCO dataset (Bitterwolf et al., 2023) into visual memory, we can remove any NINCO sample with outstanding performance on all three key unlearning metrics reported by Liu (2024): *Efficiency*: How fast is the algorithm compared to re-training? (Lightning fast.) *Model utility*: Do we harm performance on the retain data or orthogonal tasks? (Not at all.) *Forgetting quality*: How much and how well are the 'forget data' actually unlearned? (Completely and entirely.) Can machine unlearning therefore be solved with a visual memory? If the embedding model is trained on data that needs to be unlearned, machine unlearning remains challenging. If, however, the embedding model is trained on a safe, generalist dataset (e.g., a publicly available image dataset) and data that may need to be considered for unlearning later is simply put into the visual memory, then machine unlearning indeed becomes as simple as deleting a datapoint from the visual memory. This can be particularly helpful for tasks that may require private or confidential data—a model can be trained on publicly available datasets to learn general and information features and the private data can be added to a visual memory on local devices for downstream tasks to preserve privacy.

## 3.5 FLEXIBLE DATA SELECTION: MEMORY PRUNING

The ability to flexibly remove the influence of certain datapoints is not just desirable in the unlearning sense, but also advantageous in the context of dataset pruning, an emerging field that analyzes the quality of individual data points. The goal of dataset pruning is to retain only *useful* samples, while removing those that have a neutral or harmful effect on model quality. The key challenge is that in standard black-box models, it is entirely unclear whether any given sample is helpful or harmful. The gold standard is leave-one-out-training (for ImageNet, this would consist of training 1.28 million models); current methods seek to approximate this extremely costly approach with various heuristics (Feldman & Zhang, 2020; Chitta et al., 2021; Paul et al., 2021; Sorscher et al., 2022; Abbas et al., 2023a). By contrast, the contribution of a data sample to decision-making in a visual memory based system is straightforward. For any given query image $\tilde{x}$, the neighbor set

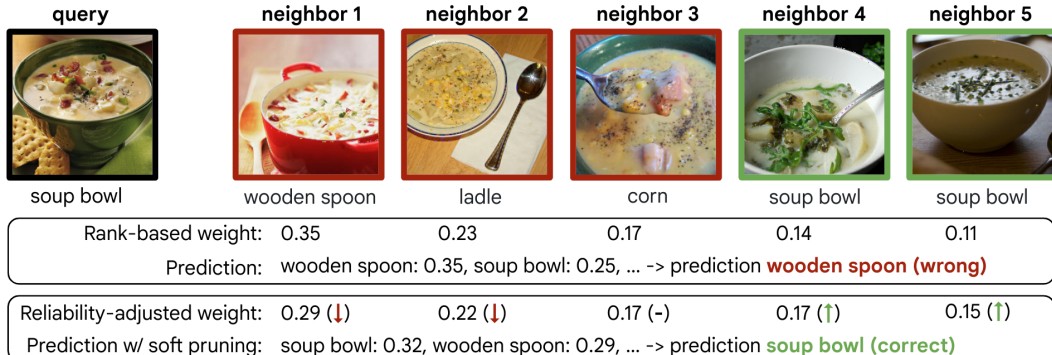

Figure 4: **Visualization of memory-based decision-making with and without memory pruning.** Given a query image, nearest neighbors are retrieved from memory via Cosine similarity in the embedding space of a model (here: five closest neighbors from the ImageNet train set, embedded via DinoV2 ViT-L14). The model's prediction is based on the weighted aggregation of the neighbor class labels. The rank-based weight decreases with the rank of the neighbor. For soft memory pruning, those weights are adjusted by the reliability of their neighbors. In the specific example here, all five neighbors appear sensible, but they have four different labels. Since the first two neighbors contributed to wrong decisions on the training set, they are downweighted via soft memory pruning, and the prediction changes to the correct class.

Neighbors($\tilde{x}$) clearly reveals which samples contributed to the decision. Furthermore, this information also highlights whether the samples were helpful (correct label) or harmful (wrong label) for the decision. We, therefore, transfer the concept of dataset pruning to memory, and propose visual *memory pruning*. To this end, we estimate sample quality by querying the ImageNet training set against a visual memory consisting of the exact same dataset (IN-train, discarding the first neighbor which is identical to the query). This approach requires no more compute than a single forward pass over the training set. We then record the number of times any given neighbor contributed to a wrong decision, resulting in a sample quality estimate. This enables us to exclude low-quality neighbors from the decision-making process by either removing them from the visual memory entirely ("hard memory pruning") or by reducing their weight compared to higher-quality neighbors ("soft memory pruning"). Method details can be found in Appendix H. In Table 3, we show that both memory pruning variants improve ImageNet validation accuracy, with soft pruning leading to larger gains than hard pruning. Figure 4 visualizes the decision-making process for a randomly selected sample where estimating sample reliability improves decision quality. Given that observing the outcome of an intervention is many orders of magnitude faster in visual memory models (as opposed to traditional leave-out-training), we are optimistic that the visual memory pruning gains we observed with two simple strategies can be improved further in the future.

Table 3: **Flexible data selection: memory pruning.** ImageNet validation accuracy improves when removing low-quality samples (hard pruning) or downweighting them (soft pruning). In contrast to standard black-box models, memory models (here: using DinoV2 ViT-L14) offer a strikingly simple way to estimate sample quality since their decisions are based on a few retrieved memory samples.

| Pruning | Plurality Voting | Distance Voting | Softmax Voting | Rank Voting |
|---|---|---|---|---|
| no pruning (standard) | 83.2 | 83.3 | 83.6 | 83.6 |
| hard pruning (ours) | 83.3 | 83.4 | 83.6 | 83.7 |
| soft pruning (ours) | **83.6** | **83.6** | **83.9** | **84.1** |

### 3.6 FLEXIBLY INCREASING DATASET GRANULARITY

In contrast to static classification, where a model is trained once without updates, a visual memory model should be able to flexibly refine its visual understanding as more information becomes available. We test this using DinoV2 ViT-L14 embeddings on the iNaturalist21 dataset (iNaturalistTeam,

2021), a large-scale imbalanced dataset of animal and plant images containing 10,000 species spanning seven taxonomic levels, from coarse (kingdom) to fine-grained (species). In a leave-one-out fashion, we simulate the discovery of a new species by putting 50 exemplars for each of the 9,999 species into memory and then step by step adding more data for the remaining "newly discovered" species—starting from zero exemplars all the way to 50 exemplars (see Algorithm 1 for an algorithmic description). In Section 3.6 we observe the following: (1.) Already before a single example of the new species is added, it can already be placed in the right part of the taxonomic tree well beyond chance (35.2% accuracy at the genus level compared to ∼0% chance). (2.) Accuracy at the species level improves substantially by adding just a handful of images of the target species (e.g., 5–10 images); a regime where training a classifier would typically fail due to data scarcity. (3.) Interestingly, adding more samples of the discovered species not only improves species-level accuracy, but also leads to a "rising tide lift" of improvements across all levels of the taxonomic hierarchy. This indicates that a visual memory is well-suited for hierarchical classification tasks and settings where data for new concepts is initially scarce but becomes more abundant over time—which is often the case in applications like fraud detection, personalized recommender systems, and scientific discovery.

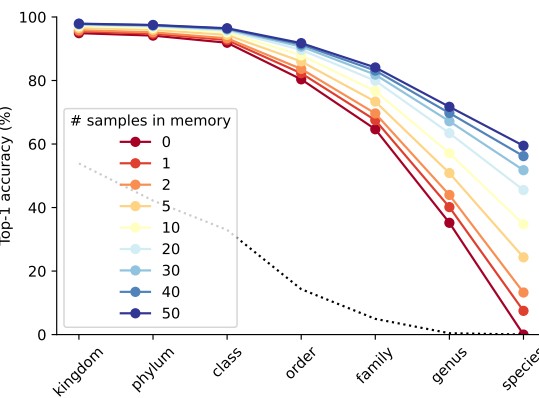

Figure 5: **Impact of memory bank size on top-1 accuracy across taxonomic levels on iNaturalist.** Top-1 accuracy for a target species across different taxonomic levels as the number of exemplars in the memory bank for that species increases from 0 to 50. Each line represents the average accuracy over all 10,000 species in the iNaturalist 2021 dataset, while the number of examples in visual memory is fixed at 50 exemplars for all other species. The black dotted line indicates baseline accuracy from predicting the majority class.

### 3.7 INTERPRETABLE & ATTRIBUTABLE DECISION-MAKING

Unlike a black-box deep learning model, a visual memory offers a natural way to understand a model's specific predictions by attributing them to training data samples (e.g. Papernot & McDaniel, 2018). In Figure 6, we visualize misclassified validation set examples from the ImageNet-A dataset (Hendrycks et al., 2021) using a memory of the ImageNet-1K training set. These randomly selected samples illustrate that many seemingly strange errors (e.g., predicting a type of fence instead of a teddy bear, or a unicycle instead of a bow tie) do in fact appear sensible given the data, raising questions about label quality of ImageNet-A—in a similar vein as label issues identified for ImageNet (Beyer et al., 2020; Shankar et al., 2020; Yun et al., 2021)—rather than about model quality. This issue is quantified in Appendix M, showing that 2 out of 5 model "errors" are instead label errors.

## 4 DISCUSSION

**Summary.** Typical neural networks are trained end-to-end: perfect for static worlds, yet cumbersome to update whenever knowledge changes. This is limiting their potential in real-world settings since the world is constantly evolving. Incorporating a visual memory, in contrast, enables a range of flexible capabilities that embrace change: lifelong learning through incorporating novel knowledge, being able to forget, remove and unlearn obsolete knowledge, flexible data selection through memory pruning, and an interpretable decision-making paradigm on which one can intervene to control its behavior. We systematically explored a simple visual memory that decomposes the task of image classification into two primitives, image *similarity* (from a pre-trained embedding representation) and *search* (via fast, scalable nearest neighbor search from a vector database). Our results demonstrate that technical improvements like RankVoting improve kNN accuracies for both DinoV2 and CLIP over the widely used SoftmaxVoting method that is sensitive to two hyperparameters (temperature $\tau$ and number of neighbors $k$). Our approach also narrows the accuracy gap between a nearest neighbor memory (best flexibility, perfect unlearning, improved interpretability) and a fixed

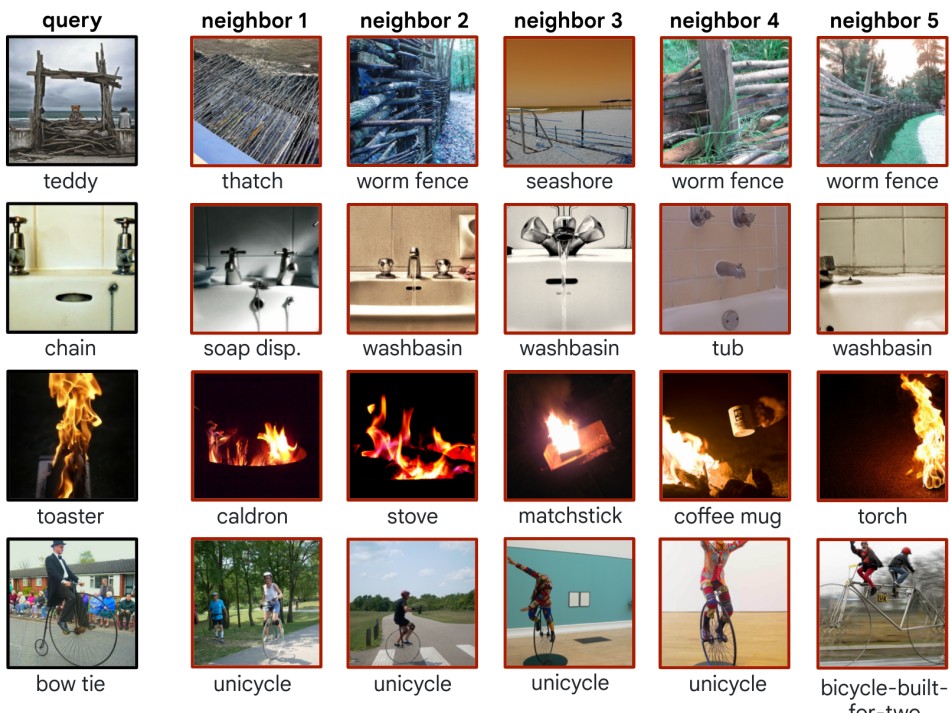

| query | neighbor 1 | neighbor 2 | neighbor 3 | neighbor 4 | neighbor 5 |
|---|---|---|---|---|---|
| teddy | thatch | worm fence | seashore | worm fence | worm fence |
| chain | soap disp. | washbasin | washbasin | tub | washbasin |
| toaster | caldron | stove | matchstick | coffee mug | torch |
| bow tie | unicycle | unicycle | unicycle | unicycle | bicycle-built-for-two |

Figure 6: **Interpretable decision-making.** A retrieval-based visual memory enables a clear visual understanding of why a model makes a certain prediction. Here, we show four randomly selected misclassified query images from ImageNet-A (Hendrycks et al., 2021) along with five nearest neighbors from DinoV2 ViT-L14 using the ImageNet-1K training set as visual memory. All labels are from the respective datasets (ImageNet-A for query and ImageNet-train for neighbors). While all neighbors visually look reasonable, not all labels do.

linear probe (highest accuracy on static image classification). More importantly, we show that visual memory enables *flexible* perceptual capabilities.

**Limitations and future work.** First, we only considered the task of image classification across a broad range of datasets. It will be interesting to extend the approach to other visual tasks, such as object detection, image segmentation, instance recognition and to image generation where a visual memory would be desirable, too (since it is prohibitively expensive to re-train large generative models every time data needs to be removed or added). Secondly, our approach relies on a fixed, pre-trained embedding model; strong distribution shifts may require updating the embedding. Self-supervised models are a particularly flexible choice, but it is an open question whether one could train smaller models that excel at their task with the help of a larger memory database. Conceptually, if a model needs to save less information in its weights, it might be possible to reduce the computational footprint of such a model. Furthermore, we sometimes observe a trade-off between flexibility and accuracy. Exploring the use of the memory pruning weights as a data selection criterion in the context of dataset pruning to improve over power-law scaling in deep learning (Sorscher et al., 2022) might be an interesting avenue for future work.

**Outlook.** Deep learning is increasingly becoming a victim of its own success: the more widely it is deployed, the stronger its limitations are felt. While the static nature of end-to-end trained networks can easily be forgotten when focusing on fixed academic benchmarks, the real world is anything but static. Data is constantly evolving, leading to the dreaded "model drift" where once-optimal models gradually become less effective (Bayram et al., 2022). Incorporating an explicit visual memory—however it may be instantiated—appears to be a promising way forward for real-world tasks where flexibility is key. While the specific approach we employ here might well be improved through more complex systems, we hope that the flexible capabilities we demonstrated might inspire and contribute to a conversation on how knowledge ought to be represented in vision models.

**Code availability.** Code to replicate experiments from this paper is available via github; for the purpose of the anonymous review period we include it as a supplementary .zip file.

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

# Appendix

We here provide the following supplemental information:

## A    AGGREGATION METHOD COMPARISON (IMAGENET-1K)

Table 4: Benchmarking different aggregation variants at different $k$ thresholds, DinoV2 ViT-L14.

| Aggregation | @10 | @20 | @30 | @40 | @50 | @60 | @70 | @80 | @90 | @100 |
|---|---|---|---|---|---|---|---|---|---|---|
| PluralityVoting | 83.2 | 82.9 | 82.6 | 82.4 | 82.1 | 82.0 | 81.8 | 81.6 | 81.5 | 81.4 |
| DistanceVoting | 83.3 | 83.0 | 82.7 | 82.4 | 82.2 | 82.1 | 81.9 | 81.7 | 81.6 | 81.5 |
| SoftmaxVoting | **83.5** | 83.5 | 83.4 | 83.3 | 83.2 | 83.1 | 83.1 | 83.0 | 82.9 | 82.9 |
| RankVoting | **83.5** | **83.6** | **83.6** | **83.5** | **83.5** | **83.4** | **83.3** | **83.3** | **83.3** | **83.3** |

Table 5: Benchmarking different aggregation variants at different $k$ thresholds, DinoV2 ViT-B14.

| Aggregation | @10 | @20 | @30 | @40 | @50 | @60 | @70 | @80 | @90 | @100 |
|---|---|---|---|---|---|---|---|---|---|---|
| PluralityVoting | 81.8 | 81.4 | 81.1 | 80.9 | 80.7 | 80.4 | 80.2 | 80.0 | 79.8 | 79.6 |
| DistanceVoting | 81.9 | 81.5 | 81.2 | 81.0 | 80.8 | 80.5 | 80.3 | 80.0 | 79.9 | 79.7 |
| SoftmaxVoting | 82.0 | 82.0 | 81.9 | 81.8 | 81.7 | 81.7 | 81.6 | 81.5 | 81.3 | 81.3 |
| RankVoting | **82.1** | **82.2** | **82.1** | **82.0** | **82.0** | **82.0** | **81.9** | **81.9** | **81.9** | **81.9** |

Table 6: Benchmarking different aggregation variants at different $k$ thresholds, DinoV2 ViT-S14.

| Aggregation | @10 | @20 | @30 | @40 | @50 | @60 | @70 | @80 | @90 | @100 |
|---|---|---|---|---|---|---|---|---|---|---|
| PluralityVoting | 78.6 | 78.2 | 77.8 | 77.4 | 77.1 | 76.8 | 76.5 | 76.3 | 76.1 | 75.9 |
| DistanceVoting | 78.8 | 78.4 | 77.9 | 77.5 | 77.2 | 76.9 | 76.6 | 76.4 | 76.2 | 76.0 |
| SoftmaxVoting | **78.9** | 78.9 | 78.7 | 78.6 | 78.5 | 78.3 | 78.1 | 78.0 | 77.9 | 77.7 |
| RankVoting | **78.9** | **79.1** | **79.0** | **78.9** | **78.9** | **78.9** | **78.9** | **78.8** | **78.8** | **78.8** |

Table 7: Benchmarking different aggregation variants at different $k$ thresholds, CLIP ViT-L14.

| Aggregation | @10 | @20 | @30 | @40 | @50 | @60 | @70 | @80 | @90 | @100 |
|---|---|---|---|---|---|---|---|---|---|---|
| PluralityVoting | 79.0 | 78.7 | 78.3 | 78.0 | 77.8 | 77.6 | 77.4 | 77.4 | 77.2 | 77.0 |
| DistanceVoting | 79.2 | 78.9 | 78.5 | 78.2 | 78.0 | 77.8 | 77.6 | 77.5 | 77.3 | 77.1 |
| SoftmaxVoting | **79.3** | 79.3 | 79.1 | 78.9 | 78.8 | 78.7 | 78.5 | 78.5 | 78.4 | 78.2 |
| RankVoting | **79.3** | **79.6** | **79.7** | **79.7** | **79.7** | **79.7** | **79.7** | **79.7** | **79.7** | **79.7** |

Table 8: Benchmarking different aggregation variants at different $k$ thresholds, CLIP ViT-B16.

| Aggregation | @10 | @20 | @30 | @40 | @50 | @60 | @70 | @80 | @90 | @100 |
|---|---|---|---|---|---|---|---|---|---|---|
| PluralityVoting | 72.8 | 72.6 | 72.3 | 72.0 | 71.7 | 71.4 | 71.2 | 70.9 | 70.8 | 70.5 |
| DistanceVoting | 73.1 | 72.9 | 72.6 | 72.3 | 71.9 | 71.6 | 71.4 | 71.1 | 70.9 | 70.6 |
| SoftmaxVoting | **73.3** | 73.3 | 73.1 | 72.9 | 72.7 | 72.5 | 72.3 | 72.1 | 71.9 | 71.7 |
| RankVoting | 73.0 | **73.7** | **73.8** | **73.8** | **73.8** | **73.8** | **73.7** | **73.7** | **73.7** | **73.7** |

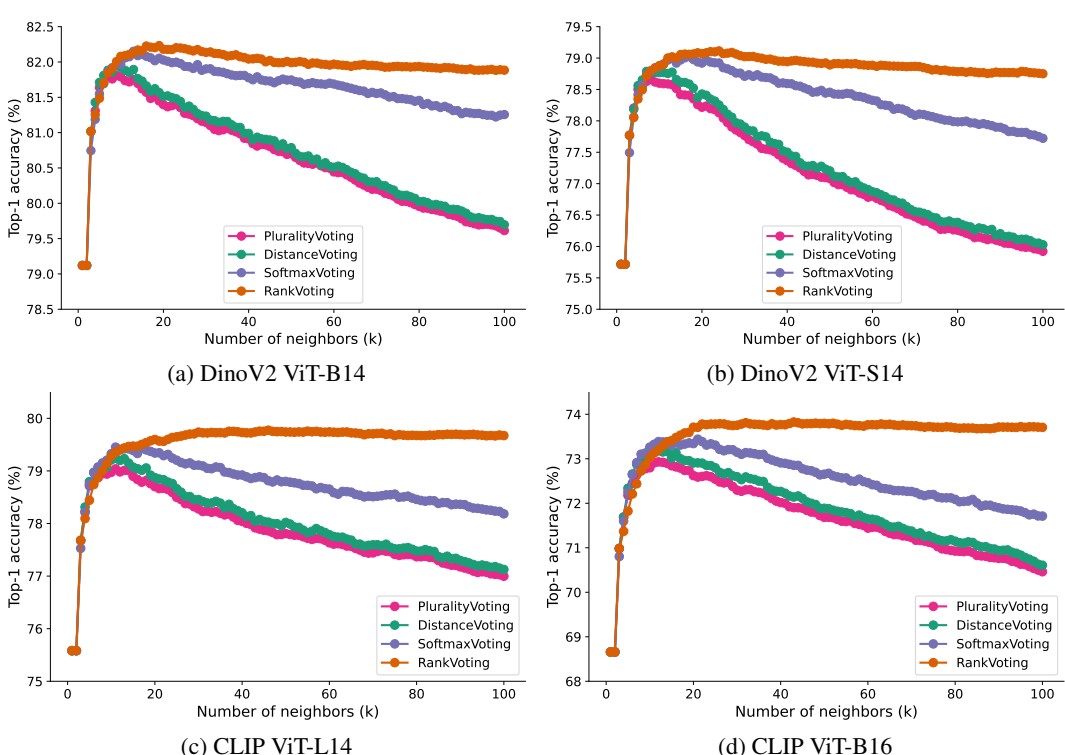

(a) DinoV2 ViT-B14      (b) DinoV2 ViT-S14

(c) CLIP ViT-L14      (d) CLIP ViT-B16

Figure 7: Aggregation method comparison on the ImageNet-1K validation set (same as Figure 2a but for other models).

Table 9: Benchmarking different aggregation variants on ImageNet-1K.

| Model | Aggegation | IN-val acc (%) |
|---|---|---|
| CLIP ViT-L14 | CLIP paper (zero-shot) | 75.3 |
| CLIP ViT-L14 | no aggregation | 76.0 |
| CLIP ViT-L14 | PluralityVoting | 79.2 |
| CLIP ViT-L14 | DistanceVoting | 79.4 |
| CLIP ViT-L14 | SoftmaxVoting | 79.6 |
| CLIP ViT-L14 | RankVoting | **79.9** |
| DinoV2 ViT-L14 | DinoV2 paper (kNN Softmax) | 83.5 |
| DinoV2 ViT-L14 | no aggregation | 81.1 |
| DinoV2 ViT-L14 | PluralityVoting | 83.2 |
| DinoV2 ViT-L14 | DistanceVoting | 83.3 |
| DinoV2 ViT-L14 | SoftmaxVoting | **83.6** |
| DinoV2 ViT-L14 | RankVoting | **83.6** |

## B    AGGREGATION METHOD COMPARISON (INATURALIST)

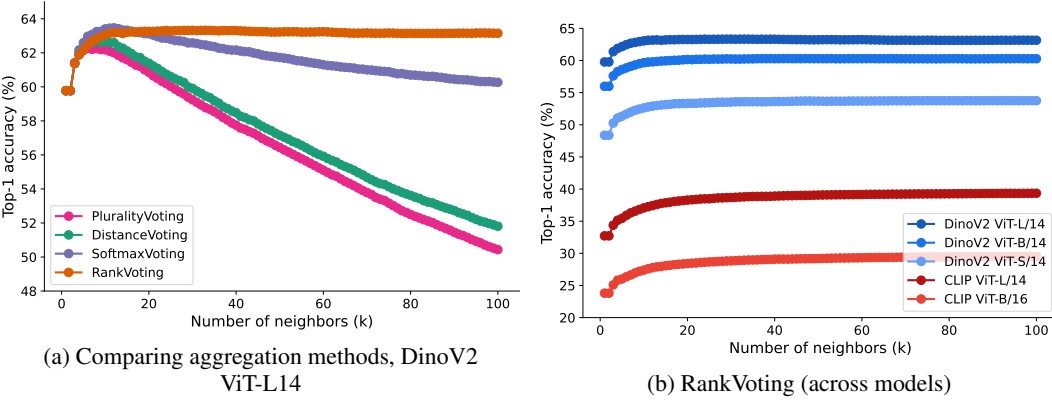

(a) Comparing aggregation methods, DinoV2 ViT-L14

(b) RankVoting (across models)

Figure 8: **Aggregating information across retrieved memory samples on iNaturalist.**  Same as Figure 2 but for iNaturalist instead of ImageNet. **(left)** Existing aggregation methods (PluralityVoting, DistanceVoting and SoftmaxVoting) are overconfident in distant neighbors, resulting in the paradox of decaying iNaturalist accuracy with more information. **(right)** This is not the case for RankVoting which shows strong and stable performance across models and choices of $k$.

## C   HYPERPARAMETER SENSITIVITY ANALYSIS

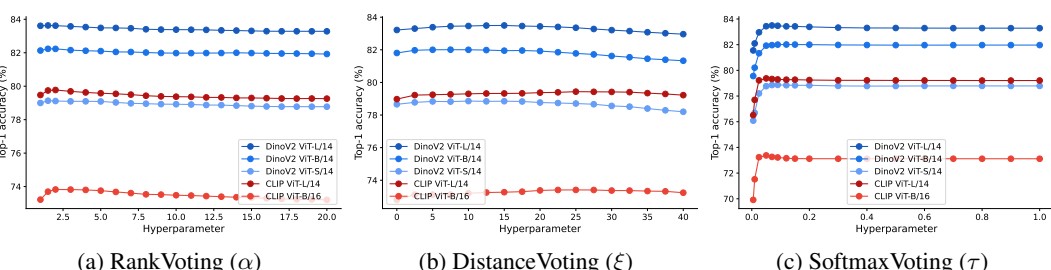

(a) RankVoting ($\alpha$)      (b) DistanceVoting ($\xi$)      (c) SoftmaxVoting ($\tau$)

Figure 9: **Sensitivity to hyperparameters for different aggregation methods.** Apart from Plurality Voting, all aggregation methods described in Section 2.2 have a hyperparameter ($\alpha$ for RankVoting, $\tau$ for SoftmaxVoting). For each model and method, we here plot the maximum performance when aggregating using a certain method, sweeping over the number of neighbors from 1 to 100, as a function of the hyperparameter. This analysis is performed to understand how sensitive the respective method is to the choice of the hyperparameter. Note that the x range is different since for instance the temperature parameter in SoftmaxVoting ranges from $[0, 1]$ while RankVoting for $\alpha = 0$ is undefined (division by zero). We therefore evaluate a broad range for each method and find that all methods have a regime in which they are relatively stable irrespective of the hyperparameter choice. Since DistanceVoting as implemented by Khandelwal et al. (2019) does not have a hyperparameter, we added a temperature-style parameter $\xi$ for the purpose of this comparison by setting $w_i = \exp\left(-\mathsf{dist}(\tilde{z}, z_{[i]})\right)^\xi$.

## D   ROBUSTNESS TOWARDS LABEL CORRUPTION

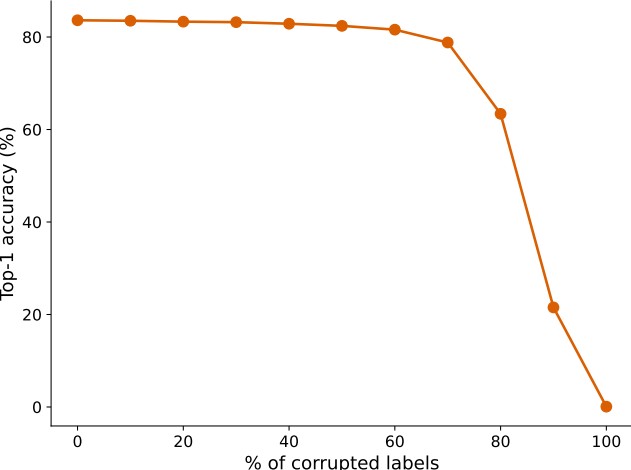

Figure 10: **Robustness towards label corruption.** How robust is a visual memory towards corrupted labels in the memory bank? This plot shows top-1 RankVoting accuracy on the ImageNet validation set as a function of how many labels in the memory (containing ImageNet-1K training set features via DinoV2 ViT-L/14) are corrupted, i.e., assigned to a random class. Intriguingly, performance stays almost unchanged all the way to about 60% (!) corrupted (random) labels in the database.

# E   HIT RATE ANALYSIS

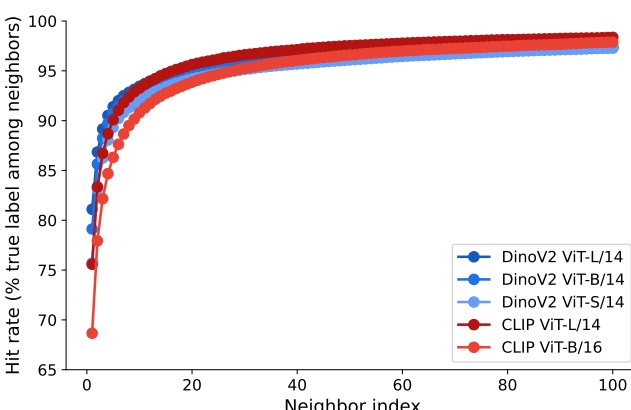

Figure 11: **Hit rate.** This plot shows the probability of the true label being contained in list of labels of the first $k$ retrieved neighbors on ImageNet-1K, for five different models and $k \in [1, 100]$. With 100 neighbors, the hit rate approaches 98% for the best model. Conceptually, this is a very high upper bound on the performance that can be achieved by a given featurizer via nearest neighbor retrieval.

# F   SCALING LAW

As we mentioned in Section 3.3, we found that a logarithmic form fits the data well between $\log_{10}$(memory size) and $\log_{10}$(error rate). Specifically, we found the following functional forms for DinoV2 ViT S14 and DinoV2 ViT L14 respectively via `np.polyfit(x, y, dim=1)`:

$$\textbf{DinoV2 ViT L14:} \quad y = -0.9434 \cdot \log_{10}(x) + 2.0704$$
$$\textbf{DinoV2 ViT S14:} \quad y = -1.0942 \cdot \log_{10}(x) + 2.3187$$

where $x = \log_{10}$(memory-size) and $y = \log_{10}$(error-rate), where memory-size $\in [10^3, 10^9]$ and error-rate in $[0, 100]$.

# G   NINCO DATASET

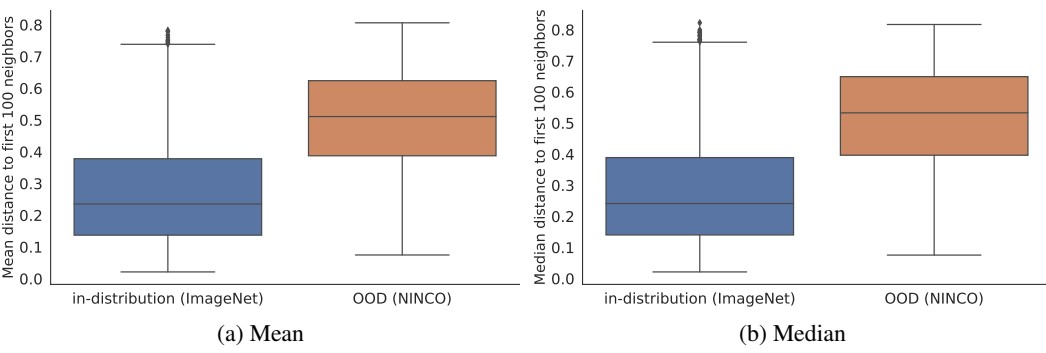

(a) Mean                                      (b) Median

Figure 12: **Distance comparison: the NINCO OOD samples are indeed out-of-distribution for the model.** In Section 3.1, we described that we can simply plug new out-of-distribution classes into memory and still perform well on both existing data as well as the new classes. This boxplot confirms that the added samples from the NINCO dataset (Bitterwolf et al., 2023) are indeed out-of-distribution for DinoV2 ViT-L14: The mean **(left)** and median **(right)** distances from query to the first 100 neighbors are substantially lower for ImageNet validation images than for OOD samples from NINCO.

Figure 12 confirms that there is a distribution difference between in-distribution data (ImageNet-1K) and OOD data (NINCO). That said, while a distribution shift exists, it is possible that individual NINCO samples were part of the training set for DinoV2. Test-set contamination is generally a concern when working with models trained on large-scale datasets, since test samples may occur as exact, semantic or near-duplicates in large training datasets (e.g. Abbas et al., 2023b). For instance, NINCO contains samples from Food-101 (Bossard et al., 2014) which are also part of LVD-142M dataset used to train DinoV2. That said, the NINCO samples belong to classes which are definitely not part of the ImageNet-train set which serves as a memory bank for our experiments, as ensured by the NINCO dataset collection process (Bitterwolf et al., 2023).

## H MEMORY PRUNING

For memory pruning from Section 3.5, we implemented two pruning methods: removing unreliable neighbors from memory entirely ("hard memory pruning"), and reducing their weight ("soft memory pruning"). We report results on the ImageNet validation set with a (potentially pruned) ImageNet-train set in memory. For hard pruning, we excluded images from memory that contributed to a wrong decision at least 128 times (this meant excluding 26,257 images for DinoV2 ViT-L14), based on querying the ImageNet-train set against a memory consisting of the ImageNet-train set and querying 100 neighbors for each sample. In order to obtain a fair comparison, instead of reporting accuracies for an arbitrary choice of $k$ (the number of neighbors) we instead evaluate accuracy for each $k$ in $[1, 100]$ and report the maximum accuracy obtained in Table 3. This ensures that differences in observed accuracy can indeed be attributed to memory pruning, as opposed to a choice of $k$. For soft pruning, instead of excluding unreliable neighbors entirely as in hard pruning, the neighbor weights (1.0 for PluralityVoting, or a rank-based weight in case of RankVoting) are instead multiplied by a reliability factor $\gamma$ with $\gamma = \frac{d}{c+v}$ where $v$ is the number of times the image contributed to a wrong decision on the ImageNet-train set, $c = 1$ to avoid division by zero, and $d = 1.75$. This results, for instance, in $\gamma = 0.88$ for images that only contribute to a single wrong decision; in $\gamma = 0.16$ for images that contribute to ten wrong decisions, and in $\gamma = 0.02$ for images that contribute to 100 wrong decisions on the training set. Images that never contributed to any wrong decision are assigned $\gamma = 1.0$, i.e. their default weight remains unchanged.

## I LINEAR PROBE DETAILS

For the linear probe results reported in the paper, we directly used the results that were reported in the DinoV2 and CLIP papers. For DinoV2, the authors froze the model backbone and trained the linear layers for 12500 iterations using SGD. Instead of training a single time, they performed a full grid search sweep over three settings (output layers in 1, 4; pooling token concatenation in yes, no, and 13 different learning rates), resulting in 52 linear probes. Then, the authors evaluated the ImageNet validation accuracy for all of those 52 probes and only reported the highest one, as described in Appendix B.3 of the DinoV2 paper. Some may call this test set tuning or double dipping; the DinoV2 paper describes it as "common practice" (Oquab et al., 2023, p. 31). CLIP linear probe results are based on a logistic regression classifier learned using scikit-learn's L-BFGS implementation, and hyperparameter sweeps are performed on a held-out set not used for evaluation, according to Radford et al. (2021).

## J LATENCY AND STORAGE

**Latency.** Nearest neighbor retrieval, fortunately, does not need to reinvent the wheel but can, instead, build on top of highly optimized workloads and libraries such as the ScaNN library (Guo et al., 2020). The ScaNN github README shows a latency comparison; with the requirement of perfect recall a million-size memory can handle roughly 500-600 queries per second.

**Storage.** In addition to latency, storage is another very practical consideration: How much does it take to store features for a large database? To put things into perspective, the ImageNet training dataset requires 154.6 GB of storage, and the ImageNet validation dataset requires 6.0 GB of storage. In comparison, as shown in Table 10, storing DinoV2 or CLIP features for the entire ImageNet

training dataset only requires between 1.9 and 4.9 GB of storage space. Thus compared to storing the training dataset, the model features account for only 1–3% of this size. This means that after constructing the memory, one may decide to keep the dataset which adds 1–3% of storage, or one may decide to delete the dataset only keeping the features which saves 97–99% of storage (compared to the dataset storage requirement). The ratio of features requiring 1–3% of the dataset size doesn't change with dataset scale since it only depends on the embedding model, thus this ratio would hold for very small datasets just as it would for a billion-scale dataset.

Table 10: **Storage requirements for ImageNet features.** Storing features in a memory database requires only about 1–3% of the space that is needed to store the dataset (154.6 GB for ImageNet-train, 6.0 GB for ImageNet-validation).

| Model | IN-train features (GB) | IN-val features (MB) |
|---|---|---|
| DinoV2 ViT-L/14 | 4.9 | 197 |
| DinoV2 ViT-B/14 | 3.7 | 148 |
| DinoV2 ViT-S/14 | 1.9 | 75 |
| CLIP ViT-L/14 | 3.7 | 148 |
| CLIP ViT-B/16 | 2.5 | 100 |

## K  HIERARCHICAL LABEL PREDICTION ALGORITHM FOR FLEXIBLY INCREASING DATASET GRANULARITY

---

**Algorithm 1** Hierarchical Label Prediction

---

**Require:** New example $x$, Hierarchical tree $T$ (with ROOT node)
1: $cur\_node \leftarrow$ ROOT
2: **for** each level $l$ in $T$ (top to bottom) **do**
3:     $candidates \leftarrow$ all_children_of($cur\_node$)
4:     $max\_p\_value \leftarrow -\infty$
5:     $label\_at\_level \leftarrow$ NULL
6:     **for** each child node $c$ in $candidates$ **do**
7:         $cross\_dist \leftarrow$ distance_distribution($x$, examples($c$))
8:         $in\_dist \leftarrow$ distance_distribution(examples($c$), examples($c$))
9:         $p\_value \leftarrow$ Kolmogorov-Smirnov test($cross\_dist, in\_dist$)
10:        **if** $p\_value > max\_p\_value$ **then**
11:            $max\_p\_value \leftarrow p\_value$
12:            $label\_at\_level \leftarrow c$
13:        **end if**
14:     **end for**
15:     $cur\_node \leftarrow label\_at\_level$
16: **end for**
17: **return** $label\_at\_level$

---

# L CALIBRATION ANALYSIS

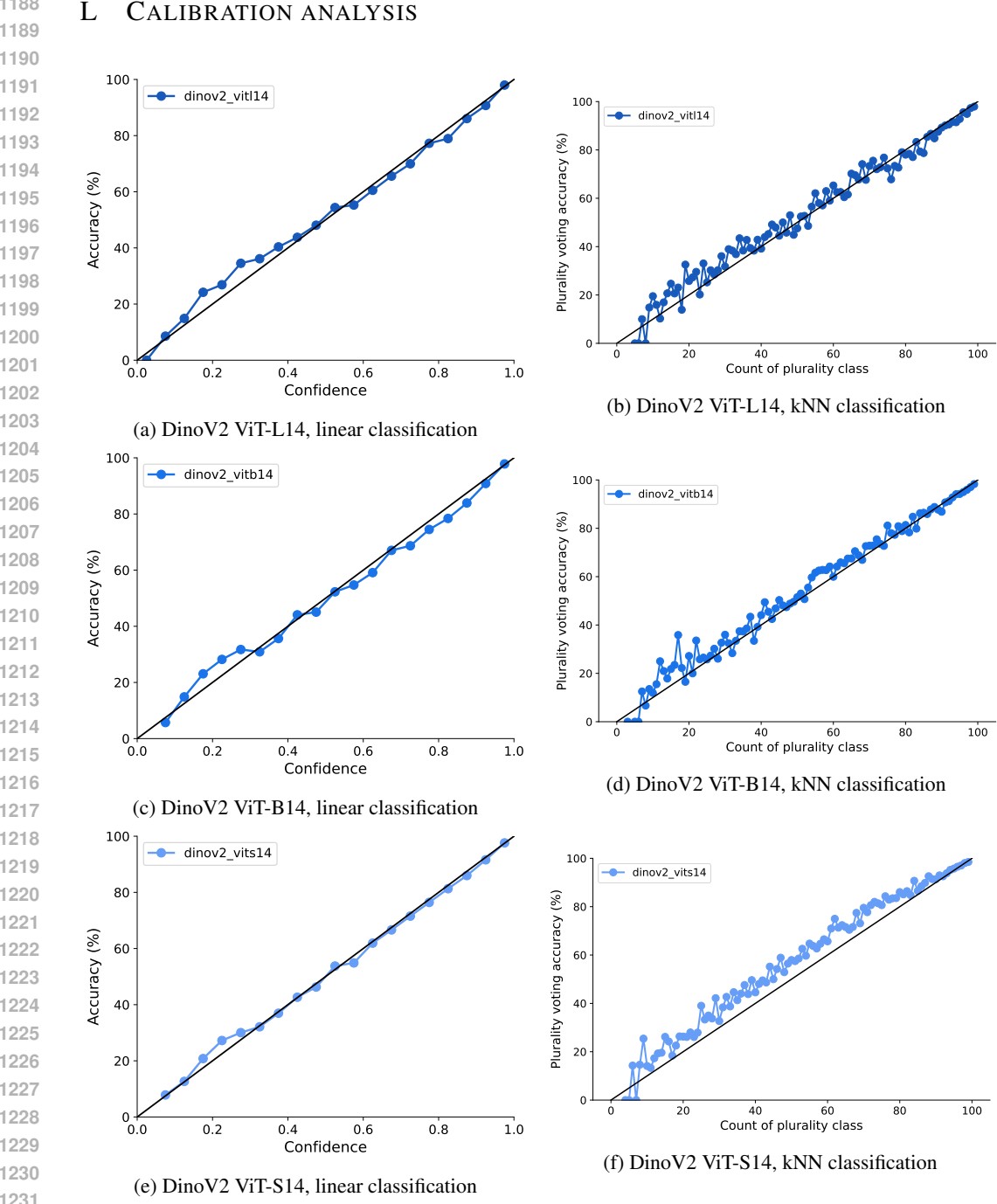

(a) DinoV2 ViT-L14, linear classification

(b) DinoV2 ViT-L14, kNN classification

(c) DinoV2 ViT-B14, linear classification

(d) DinoV2 ViT-B14, kNN classification

(e) DinoV2 ViT-S14, linear classification

(f) DinoV2 ViT-S14, kNN classification

Figure 13: **How well are predictions calibrated?** Left column: Accuracy vs. confidence from Softmax of linear classifier for three DinoV2 variants. Right column: Accuracy vs. count of plurality class among first 100 neighbors for the same three DinoV2 variants. A DinoV2-based kNN classifier is well calibrated, as is the DinoV2 softmax.

# M IMAGENET-A ERROR ANALYSIS

As shown in Figure 6, many "errors" on ImageNet-A appear to be perfectly reasonable predictions that are caused by dataset label issues as opposed to model mistakes. More randomly selected ImageNet-A samples, along with nearest neighbors, are shown in Figure 14. To quantify the issue,

we performed a human experiment on a randomly selected subset of ImageNet-A images (N=100) where the dataset label and the prediction from DinoV2 ViT-L14 with JFT memory disagree. We presented the image alongside the original ImageNet-A label and our model-predicted label to three human observers, asking them to identify which of the labels best describes the image (of course, without telling them which of the labels is the dataset label). The result was that in 39.3% (!) of cases (std: $\pm 1.25\%$), the DinoV2 label was assessed as being better/more suitable than the original dataset label—i.e., roughly 2 out of 5 model "errors" are in fact dataset label errors, quantifying the ImageNet-A label quality issue we alluded to in Figure 6. This percentage can be used to estimate how correcting problematic labels influences performance. Instead of the original model's 61.1% accuracy on ImageNet-A, due to label errors the 'corrected' accuracy is instead 76.4% (a delta of $+15.3\%$ in absolute terms or $+25.0\%$ in relative terms).

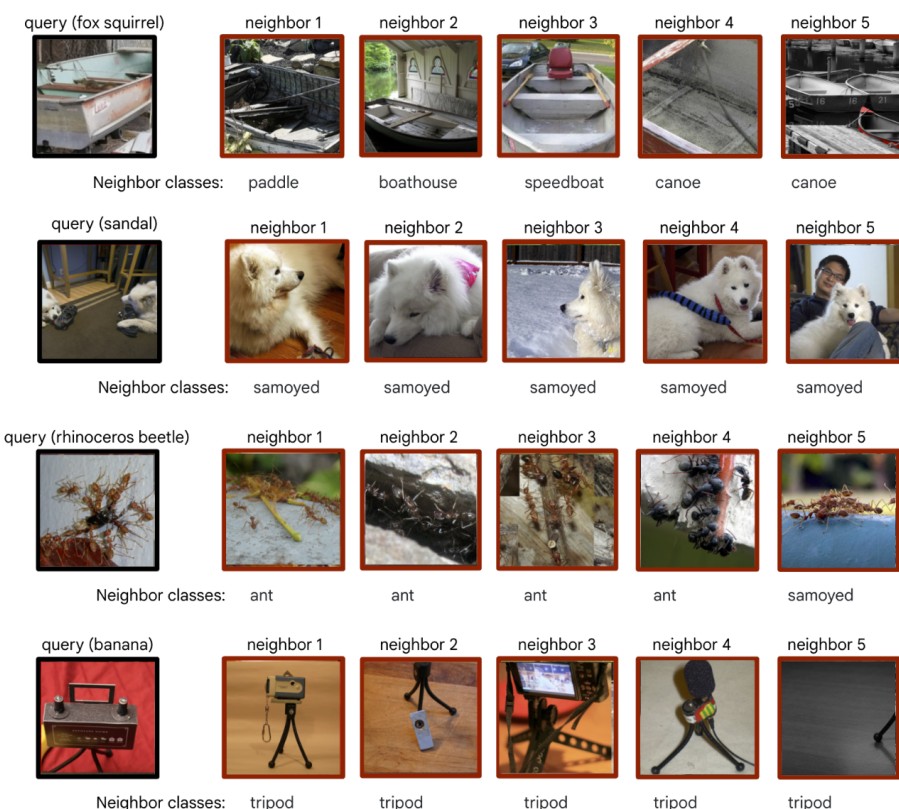

Figure 14: **Interpretable decision-making.** A retrieval-based visual memory enables a clear visual understanding of why a model makes a certain prediction. Here, we show four randomly selected misclassified query images from ImageNet-A (Hendrycks et al., 2021) along with five nearest neighbors from DinoV2 ViT-L14 using the ImageNet-1K training set as visual memory. All labels are from the respective datasets (ImageNet-A for query and ImageNet-train for neighbors). While all neighbors visually look reasonable, not all labels do.

## N    COMPOSITIONALITY ANALYSIS

A flexible visual memory also provides a path to analyze representations of various models, particularly, how different models represent multiple concepts in an image. We study this for an ImageNet-train visual memory of DinoV2 ViT-L14 and CLIP ViT-L14. We use manually selected query images from outside the ImageNet dataset that have multiple objects from the ImageNet labels. We query the visual memory for nearest neighbors of the query image. Subsequently, we obtain the *residual image* by subtracting the features of the nearest neighbor from the features of the query image. We, then, obtain the nearest neighbors for the residual image from the visual memory. We plot the

results in Figure 15 which shows that DinoV2 ViT-L14 and CLIP ViT-L14 represent concepts in their features in a different manner. The nearest neighbors for DinoV2 are mostly images with a single concept (or object) from the query image. The residual image, subsequently, leads to nearest neighbors dominated by another single object in the query image. In contrast, CLIP often results in nearest neighbors that are generally a blend of concepts from the query image. These qualitative explorations are simple demonstrations of the advantages of an interpretable decision-making process provided by a flexible visual memory.

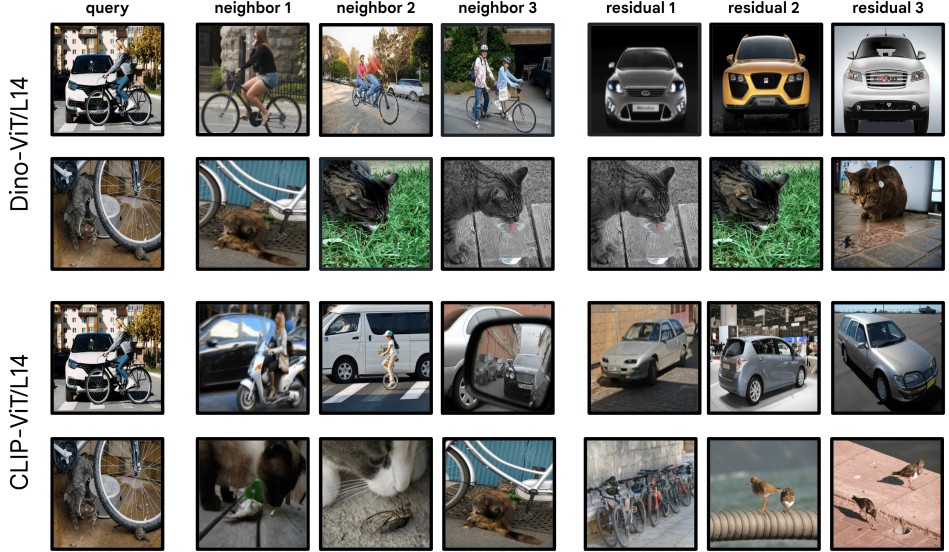

Figure 15: **Compositionality of representations.** The first column indicates a query image; the next three columns are the three nearest neighbors from the training set. The last three columns are the *residual* images, obtained by subtracting the features of the nearest neighbor (2nd column from the left) from the features of the query image (1st column from the left). The nearest neighbors for DinoV2 are mostly images with a single concept (or object) from the query image. The residual image, subsequently, leads to nearest neighbors dominated by another single object in the query image. In contrast, CLIP often finds neighbors that are a blend of concepts from the query image.

