# OpenReview forum: "Towards flexible perception with visual memory"
_ICLR.cc/2025/Conference — Submitted to ICLR 2025_

### Official Review · Reviewer_X1Pn · 2024-10-15

**Soundness:** 3
**Presentation:** 4
**Contribution:** 3
**Rating:** 6
**Confidence:** 3

**Summary:**

This paper introduces a flexible and interpretable visual memory system that integrates deep neural networks with a database-like structure to improve image classification. By separating knowledge representation from the model’s weights, the system enables easy addition, removal, and modification of data without retraining, addressing the limitations of static knowledge in traditional models. Key features include scalable data management, machine unlearning, and transparent decision-making processes. The system uses a k-nearest neighbour retrieval mechanism and RankVoting to enhance sample aggregation for improved performance, highlighting the potential of using visual memory in deep learning models to adapt to dynamic data and evolving knowledge.

**Strengths:**

1. The paper is well-written and easy to follow.
2. Investigation into different voting strategies looks interesting.
3. The idea of using visual memory database + retrieval is a promising strategy for visual recognition, given its flexibility, OOD robustness, and adaptation capacity to ever-growing concepts.

**Weaknesses:**

1. One limitation of retrieval and search engines is that the cost of storage and inference increases and the accuracy of recognition decreases as the memory database size grows. Although computing budget and accuracy can be traded off, this remains a limitation. Additionally, regarding the computation cost, I wonder if there is a rough estimate of the inference time when the database size increases to 1 billion images.
2. The adaptation capacity of the proposed visual memory to accommodate ever-growing concepts assumes that the image encoder can produce meaningful embeddings for new concepts. While for geometrically distinctive concepts, I believe this is less of a concern for DINO representations, as they are observed to contain rich geometric information, for concepts where class label correlates more with semantics rather than geometry, I wonder if the adaptation capacity still holds.

**Questions:**

The result of using Gemini re-ranking is quite strong and interesting. I wonder if this is due to VLM's in-context learning capacity. A discussion on this can be helpful.

**Details Of Ethics Concerns:**

I don't have any prominent ethics concern

---

> ### Author Response · Authors · 2024-11-20
> **Author response**
>
> Dear Reviewer X1Pn,
>
> Thank you for your positive review, we’re happy to hear that you found the paper **“well-written”** with **“excellent presentation”**, and appreciated the approach as a **“promising strategy for visual recognition”** highlighting the **“potential of using visual memory in deep learning models to adapt to dynamic data and evolving knowledge”**, with **“strong and interesting”** Gemini re-ranking results.
>
> Please find detailed answers to your questions below.
>
> *One limitation is that … the accuracy of recognition decreases as the memory database size grows*
> This might be a misunderstanding. The scaling analysis from Figure 3 shows that the error rate decreases as the memory size grows, i.e. we actually observe increasing accuracy with scale - thus this is not a limitation but an advantage. Apologies if that wasn’t clear; we now added “i.e., recognition accuracy increases with memory scale” to the figure caption.
>
> *The cost of storage and inference increases with memory scale / regarding the computation cost, I wonder if there is a rough estimate of the inference time when the database size increases to 1 billion images.*
> Great question. Please see https://openreview.net/forum?id=HoyKFRhwMS&noteId=YIxTk8VKZg for storage & latency estimates; we're also working to get a plot of retrieval latency as a function of memory scale which we will add.
>
> *Dino representations contain rich geometric information; but for concepts where class label correlates more with semantics rather than geometry, I wonder if the adaptation capacity still holds.*
> This is an interesting point. One could, for instance, think of a case where a model is tasked to retrieve “smiling cats” but not “sad cats” as an example of a semantic (rather than visual/geometric) similarity. While our current work focuses on image encoders with strong visual embeddings like DinoV2, the memory framework itself is agnostic to the encoder used: for use cases where visual similarity matters more, current encoders work well; for cases where semantic similarity (like smiling/sad cats) is more important, future encoders that may better capture semantic information could readily be integrated into our system. Presumably, the only case that wouldn’t work well is expecting a model to handle both semantic and visual similarity at the same time, without telling the model which of these similarity measures to prioritize in case they disagree.
>
>
> *The result of using Gemini re-ranking is quite strong and interesting. I wonder if this is due to VLM's in-context learning capacity. A discussion on this can be helpful.*
> Great question. We believe it is indeed the case that this strong result is due to Gemini’s in-context learning capacity, since the resulting predictions from Gemini re-ranking are better than Dino nearest neighbor predictions (83.6% on IN-val), and also better than Gemini’s predictions when no in-context neighbors are provided (69.6%). In fact, using only 10 in-context neighbors improves Gemini’s accuracy from 69.6% to 87.4%, and using 50 in-context neighbors leads to 88.5% accuracy. We now explicitly mention this connection to in-context learning (added to the paragraph on Gemini re-ranking at the end of Section 2.2).
>
> Kindly let us know whether we have addressed your questions.

---

> > ### Comment · Reviewer_X1Pn · 2024-11-21
> >
> > Thank you to the authors for their detailed response and clarifications. The additional explanations have sufficiently addressed my concerns. I agree with the argument regarding the encoder-agnostic memory framework, and I am inclined to support its acceptance.

---

> > > ### Author Response · Authors · 2024-11-22
> > > **Thanks!**
> > >
> > > Thanks for taking the time to get back to us, we're glad to hear we were able to address your concerns!

---

### Official Review · Reviewer_AsSs · 2024-10-28

**Soundness:** 2
**Presentation:** 3
**Contribution:** 1
**Rating:** 3
**Confidence:** 5

**Summary:**

This paper makes the argument that performing classification through nearest-neighbour search (in an embedding space) has many more desirable properties than more common approaches to classification found today - namely: that its much easier to add new data (and new classes) as this doesn't require retraining, that things (both images and classes) can easily be unlearned/removed, and the decisions might be more interpretable. The authors demonstrate that embeddings from recent pre-trained (vision-language) models are effective for use in a knn context and with appropriate weighting functions can outperform linear probing.

**Strengths:**

Overall, the paper is well written and easy to read if somewhat whimsical in style. The paper does highlight well some of the problems with current approaches to image classification, particularly in the context of deployment of models where the training data might change (for various reasons) regularly. The proposed approach of using weighted KNN seems to work reasonably, and the use of a scalable NN search engine helps with scaling to large numbers of instances in the memory.

A wide range of different aggregation methods are tested, and there are comparisons against standard linear probing baselines. I like the experiments that look at OOD performance and this is particularly interesting.

**Weaknesses:**

I think the biggest problem I have is that the contribution of the paper is not really clear - almost everything presented is rather obvious and based on well studied approaches; use of NN methods for classifying images based on features dates at least back to the days of Turk & Pentland (https://sites.cs.ucsb.edu/~mturk/Papers/mturk-CVPR91.pdf) in 1991 (if not before), and research on fast nearest neighbour approaches for solving KNN problems based on image features also has a long history (e.g. Sivic's VideoGoogle from the early 2000s was using inverted indexes for object matching/detection - https://www.robots.ox.ac.uk/~vgg/publications/2003/Sivic03/sivic03.pdf). Similarly, there is much research on KNN aggregation methods, including many that utilise rank (e.g. http://www.jcomputers.us/vol6/jcp0605-01.pdf & there are even articles suggesting reciprocal rank: https://visualstudiomagazine.com/articles/2019/04/01/weighted-k-nn-classification.aspx), so the contribution of the proposed RankVoting scheme is not clear. That new "memories" can be added/removed seems "obvious".

Experimentally, I feel there are quite a few problems. Firstly, it's not clear how the extreme class-imbalance in datasets like imagenet are taken into account; its non-obvious to me that it even makes sense to use a fixed value of $k$ in such cases. Secondly, Many of the presented results also do not show a clear win for any particular method (e.g. table 1 - difficult to claim RankVoting is better than softmax; table 2 - quite a mixed bag).

In the cases where a linear probe has been trained its not clear what procedure was adopted and how optimal the learned decision boundary is; was this a linear SVM or just a linear layer trained with gradient descent? does that not make a difference?

The pruning experiments are interesting, but are they not rather highlighting problems with the data (labels)? Presumably, more generally, this could be an approach for improving data quality rather than specifically improving the approach presented (although of course if does reduce computational complexity).

The paper doesn't seem to give any thought to the complexity of the proposed approaches - how big is the database of embeddings? how long does retrieval take? How does this compare to state of the art classification models (e.g. inference time, and perhaps comparing number of model weights to feature-extractor model weights + database size)?

**Questions:**

As per my comment above, have the authors considered the effect of class-imbalance in the training data and what this means with respect to $k$ and potential accuracy? When you get improvements in performance where do these come from?

What is the set-up for the linear probes? How was this validated?

Can you comment on the computational efficiency aspects? how does it scale with instances? How long does a query take? How much storage is required?

---

> ### Author Response · Authors · 2024-11-20
> **Author response (1/2)**
>
> Dear Reviewer AsSs,
>
> Thank you for taking the time to review our manuscript. We’re happy to hear that you **“like the experiments that look at OOD performance”**, found the paper **“well written”** and believe that it **“does highlight well some of the problems with current approaches to image classification”**.
>
> *Linear probe setup: What is the set-up for the linear probes? How was this validated?*
> In order to avoid the possibility of comparing against a straw man / weak baseline, we directly used the linear probe results that were reported in the DinoV2 and CLIP papers - this wasn’t clear from the manuscript but we have now added this information. For DinoV2, the authors froze the model backbone and trained the linear layers for 12500 iterations using SGD. Instead of training a single time, they performed a full grid search sweep over three settings (output layers in {1, 4}; pooling token concatenation in {yes, no}, and 13 different learning rates), resulting in 52 linear probes. Then, the authors evaluated the ImageNet validation accuracy for all of those 52 probes and only reported the highest one, as described in Appendix B.3 of the DinoV2 paper. Some call this test set tuning or double dipping; the DinoV2 authors call it “common practice”. In contrast, the CLIP results are based on a logistic regression classifier learned using scikit-learn’s L-BFGS implementation, and hyperparameter sweeps are performed on a held-out set not used for evaluation, according to the CLIP paper. We now added a section to Appendix I (“linear probe details”) explaining the setup and link it from Table 2.
>
> *Complexity and computational efficiency: How does it scale? How long does a query take? How much storage is required?*
> Those are excellent questions. Please see https://openreview.net/forum?id=HoyKFRhwMS&noteId=YIxTk8VKZg for storage & latency estimates.
>
> *Does it make sense to use a fixed value of k given class imbalance / the effect of class-imbalance in the training data and what this means with respect to k and potential accuracy?*
> That’s an important consideration. First of all, we fully agree that it is important to develop a system that works well if we don’t know the optimal fixed value of k, and it would be bad if the system suffered from poor performance in the case of imbalanced data. We therefore analyzed whether class-imbalance has an effect on ImageNet-1K validation accuracy (when using ImageNet-1K train features from DinoV2 ViT-L/14 in in memory). The result is visualized here: https://ibb.co/t2Bj5BB. While the accuracy of a class varies depending on the class difficulty, we do not observe a systematic disadvantage for classes with fewer images, suggesting that class-imbalance is not an issue here. This pattern holds irrespective of k, at least for the range of k within [1, 100] that we consider for ImageNet. Naturally, it is not advisable to choose k > num_images_per_class (for ImageNet, this would mean k > 732).
>
> *Many of the presented results also do not show a clear win for any particular method.*
> Tables 4, 5, 6, 7 and 8 show very systematic wins for RankVoting; they were moved to the appendix for space reasons. As you mention in your review, choosing a fixed k is often a bit of an arbitrary choice. While we cannot choose k flexibly depending on the class - if we did, we would have to know the class in advance - we believe it’s important to have methods that generalize across different choices for k, which is what RankVoting achieves. SoftmaxVoting precisely only on par with RankVoting if we know which k to pick - which is hard to know a priori. Existing SSL literature sometimes treats k as an hyperparameter which is optimized / swept over (e.g. DinoV2 evaluates a range of different settings - on the ImageNet validation set which is also used to report final performance), which can be avoided since RankVoting is very stable and leads to high performance across different choices of k (as shown in Tables 4–8).

---

> ### Author Response · Authors · 2024-11-20
> **Author response (2/2)**
>
> *The contribution of the paper is not really clear (long history of using nearest neighbor methods for image classification; a lot of research on aggregation methods)*
> First of all - thanks for the additional references; we have added them to our history & related work section in the introduction. We entirely agree with the reviewer that there is a long history of using kNN for core machine learning tasks. In many ways, our goal is to remind the deep learning field - which sometimes has a tendency to be myopic - that there is a host of well established techniques that could be applied to address core and central challenges in deep learning. Our work asks how such well established techniques work in a modern deep learning setting, specifically for building a visual memory. We demonstrate that one can address many of these core challenges while also providing a substantial boost in terms of a performant and practical system. For example, deep neural networks have a static knowledge representation that is hard to update, hard to unlearn, and it is hard to understand how a decision is made. We built a working proof-of-concept alternative: By building on the long history of fast nearest neighbor methods, and “marrying” them with a powerful deep learning representation (SSL features from DinoV2) and a billion-scale visual memory. Thus our main message isn’t “use RankVoting instead of SoftmaxVoting” - that’s just a technical choice. **Our main message could be summarized as “Deep learning should have a flexible visual memory, because this could solve many of its problems (static knowledge, hard to update, hard to unlearn, hard to understand how a decision is made).”** Our paper then experimentally demonstrates the potential of a flexible memory for those challenges. As we note in the outlook, “while the specific approach we employ here might well be improved [...], we hope that the flexible capabilities we demonstrated might inspire and contribute to a conversation on how knowledge ought to be represented in vision models.”
> In response to your feedback - that the main contribution needs to be articulated more clearly - we have made the following changes to the paper:
> 1. Expanding related work as described above.
> 2. Toning down description of RankVoting, instead of writing “we invent RankVoting” we note that this is similar (albeit not identical to) the approach of Guo et al. 2011 in a non-deep learning context and rephrase as: we propose using RankVoting as an alternative to the widely used SoftmaxVoting method that is considered the SOTA method in deep learning and e.g. employed by common SSL models.
> 3. Clarifying main goal by adding paragraph to end of introduction: “We argue that the way current deep learning models represent knowledge (static knowledge representation, hard to update, hard to unlearn, hard to understand how a decision is made) is problematic. As an alternative, we built a working proof-of-concept: By building on the long history of nearest neighbor methods, and “marrying” them with a powerful deep learning representation (such as SSL features from DinoV2) and a billion-scale visual memory.
>
> Does that address your concern?
>
> *That new “memories” can be added/removed seems “obvious”*
> We agree that unlearning in the context of a nearest neighbor visual memory is straightforward, but in deep learning the unlearning problem is considered a major challenge as e.g. evidenced by the NeurIPS unlearning competition last year - https://unlearning-challenge.github.io/ - with 5K registrations and 1.9K submissions. For this reason, we believe that this may be an important opportunity for deep learning to take inspiration from the retrieval literature to build a flexible visual memory, enabling unlearning as described in Section 3.4.
>
> *The pruning experiments are interesting, but are they not rather highlighting problems with the data (labels)? Presumably, more generally, this could be an approach for improving data quality rather than specifically improving the approach presented (although of course if does reduce computational complexity).*
> We’re glad to hear you appreciated the pruning experiments. Indeed, a valuable byproduct of memory pruning is its potential for improving data quality. One possibility would be to use the pruning weights obtained from memory pruning as a data selection criterion in the context of dataset pruning, where recent work has shown that given a sufficiently powerful data selection metric it is possible to improve over power-law scaling (https://arxiv.org/pdf/2206.14486). We have this potential future direction to the manuscript.
>
> Kindly let us know whether we have addressed your concerns / answered your questions.

---

> > ### Comment · Reviewer_AsSs · 2024-11-22
> >
> > I'd like to thank the authors for taking the time to respond.
> >
> > I feel that the arguments that the deep learning field can be "myopic", and that the takeaway message should be that something like a "visual memory" might address some existing clear problems are reasonable ones, and particularly agree with the first one. However, I still do not understand what the actual contribution of this paper is and would ask the authors to specify more clearly; if the intent is to convey that a visual memory with KNN and deep-learning based features can be used to solve problems, then has that not already been shown by Nakata et al? If the intent is to show that features from particular representational spaces from modern self-supervised models like DINO can be used in this context, then wasn't that already shown by Caron et al? If the intent was to demonstrate that the proposed approach can "unlearn", than is it not already widely known that features can be removed from indexing structures used for nearest-neighbours?
> >
> > If the intention of the paper was to combine concepts that already exist, then this is okay, but needs to be much more honestly reflected in the contributions. As it stands at the moment the claimed contributions are being considerably oversold.
> >
> > My overall feeling is that the paper needs at least to to be rewritten to make clear the concrete contribution(s) in the context of existing literature. This would not only in terms of the myriad of approaches that have used KNN for (image) classification (including previous approaches which have attempted to marry deep learning and KNN - e.g. https://arxiv.org/abs/1803.04765), but also the long history of vector indexing techniques for visual features, and also in the context of research on visual memory, both within deep learning and outside. Whilst the related work section of the paper references many of these other attempts it feels very superficial in terms of truely positioning the contributions of this work scientifically or conceptually. If the goal of the paper is to be about the engineering aspects of building a system that marrys together different components to achieve a goal, then that is fine although that should be made clear, and the focus should perhaps shift more towards what happens with different design choices.

---

> > > ### Author Response · Authors · 2024-11-25
> > >
> > > Thank you very much for taking the time to reply and explain your thoughts! It seems that there are two main suggestions: To clarify/explain the paper’s contributions, in particular in relation to other work; and to address what is perceived as overselling claims or contributions.
> > >
> > > Regarding the contributions of this paper:
> > > As mentioned, the goal of this article is to argue that deep learning should have a flexible visual memory, because this could solve many of its problems (static knowledge, hard to update, hard to unlearn, hard to understand how a decision is made).
> > > In order to make progress towards realizing this goal, we contribute the following technical advances:
> > > 1. Showing that **“simple scales well”**: a conceptually simple visual memory, using established building blocks from the literature like SSL features, kNN classification and scalable search, scales to the billion-scale regime in a performant way. This is particularly relevant given that e.g. Nakata et al only considered three orders of magnitude fewer data (ImageNet); their technical approach does not scale much further since it requires loading both the database and the query set into GPU memory which is infeasible for larger datasets. Additionally, in order to improve simplicity, we propose using RankVoting which reduces the need to carefully select hyperparameter k (as is common practice e.g. in the SSL kNN literature).
> > > 2. Demonstrating the promise of a visual memory in terms of the **flexible capabilities** it enables - *unlearning* with perfect guarantees which is currently considered a major open problem in deep learning, *controlling sample influence* through memory pruning, and an *improved understanding of how decisions relate to datapoints* (interpretable decision-mechanism). These capabilities are, in the words of other reviewers, “addressing practical needs” (S9w1), “a promising strategy for visual recognition given its flexibility and adaptation capacity” (X1Pn), and “not only provide a practical improvement but also stimulate important discourse that could shape future research directions” (pVvh).
> > >
> > > We’d be happy to sharpen the description of these contributions, and in particular make it crystal clear that the building blocks of our simple visual memory (SSL features + kNN classification + scalable vector search) are well established in the literature and should by no means be considered a contribution of this work. Would that address your concern of overselling and situating the paper in the context of related work? (In the event that this is not the case, we’d appreciate it if you could point us to specific sentences/paragraphs that you’re perceiving as overselling claims or contributions.)

---

> > > > ### Comment · Reviewer_AsSs · 2024-11-28
> > > >
> > > > I'd once again like to thank the authors for taking the time to respond. The last set of comments does help better set the scene for the paper, and I would encourage the authors to rework the text of the paper to make these arguments much more explicitly. I would also really like to encourage the authors to better think about the novelty of their work in the context of what has already been done / is known (for example, the "simple scales well" claim => https://arxiv.org/pdf/1702.08734 did billion-scale k-nearest-neighbour search with image features in 2017). When making these adjustments I would also strongly suggest that the authors think about how they reflect on the claims they make in a more scientific manner (claims like the algorithm being "Lightning fast." (line 356-357) are not statements with any scientific meaning).
> > > >
> > > > I am hopeful that authors will be able to make changes to improve the positioning of their work in an honest way. I note however that I am still very uncomfortable with the novelty of the contributions.

---

> > > > > ### Author Response · Authors · 2024-11-29
> > > > >
> > > > > Thanks for your response. We're glad to hear that the clarification improves "setting the scene for the paper", and we're committed to updating the writing to reflect this for the camera ready version. We will also give the paper another read to check for statements that would benefit from quantifying to make them more scientific.
> > > > >
> > > > > (Your comment regarding the efficiency of unlearning being 'lightning fast' is well taken; we're happy to quantify and rephrase this statement. It refers to the speed of removing an image from a visual memory database through a 'removal' operation. Deleting an image from disk takes about two hundredths of a second.)

---

### Official Review · Reviewer_S9w1 · 2024-11-01

**Soundness:** 3
**Presentation:** 3
**Contribution:** 3
**Rating:** 6
**Confidence:** 5

**Summary:**

The paper introduces a "visual memory" framework for image classification, aiming to make perception more adaptable by decoupling representation from memory. By combining image similarity from pretrained embeddings with fast nearest-neighbor retrieval, this framework allows flexible handling of knowledge updates, enabling data addition, memory pruning, and unlearning. The proposed RankVoting aggregation method surpasses previous methods and achieves high accuracy across ImageNet and billion-scale datasets. The work illustrates the potential of visual memory in achieving continual learning without re-training while supporting interpretability and control in deep learning models.

**Strengths:**

1. Combines the strengths of deep neural networks with database principles, enabling flexible memory operations such as adding, removing, and unlearning data at scale.
2. Achieves high accuracy and flexibility, outperforming baseline voting methods, especially with RankVoting and Gemini-based reranking.
3. Supports continual learning, efficient memory management, and interpretability, addressing practical needs like model drift and data privacy through machine unlearning.

**Weaknesses:**

1. Limited application focus on other visual tasks (e.g., object detection, segmentation) beyond classification. How might the visual memory approach be adapted for tasks like object detection or segmentation? What unique challenges would arise in those domains?
2. Reliance on a fixed pretrained embedding model may limit adaptability under significant distribution shifts. Have you considered approaches to update the embedding model incrementally as new data becomes available? This could help address potential limitations in adaptability under distribution shifts.
3. Sensitivity of SoftmaxVoting to hyperparameters could impact performance stability.
4. Limited ablation studies on the impact of visual memory size across various neural network architectures.
5. No detailed analysis on memory retrieval latency and computational overhead for large-scale memory databases. Could you provide more information on the memory retrieval latency and computational overhead, especially for the billion-scale experiments? This would help readers understand the practical implications of scaling up the visual memory approach.
6. Lack of performance benchmarking across diverse dataset distributions.
7. Absence of discussions on handling adversarial attacks or noise within the visual memory framework. How robust is the visual memory approach to adversarial examples or noisy data? It would be valuable to see an analysis of how the system performs when the memory contains corrupted or adversarially crafted samples.
8. Limited interpretability assessment for users to control specific decision boundaries. Could you provide more details on how users might interact with or control the decision-making process in your visual memory framework? Specifically, how might one adjust decision boundaries or influence the model's predictions in practice?
9. Potential limitations in scalability for different embedding models with high memory footprints.
10. Insufficient exploration of non-image applications (e.g., text-based memory) within the flexible memory paradigm.

**Questions:**

1. Could the framework extend to tasks like segmentation or object detection, and if so, how would visual memory be adapted?
2. How does RankVoting handle adversarial or noisy samples in memory retrieval? Would further methods mitigate this?
3. Would memory retrieval latency impact real-time performance, especially in billion-scale databases?
4. What would be the computational cost for re-ranking with Gemini, particularly on constrained devices?
5. Could the proposed system be combined with incremental learning techniques to improve distribution shift resilience?
6. How might RankVoting's performance vary with different embedding models?
7. What measures could prevent sensitive or incorrect data from being memorized, impacting model reliability?
8. Are there optimizations to manage storage costs of billion-scale datasets without affecting retrieval accuracy?
9. Could memory pruning methods be customized for specific tasks, improving model performance further?
10. How does the model handle cases where similar but incorrect neighbors dominate the nearest neighbors?

---

> ### Author Response · Authors · 2024-11-20
> **Author response (1/2)**
>
> Dear Reviewer S9w1,
>
> Thank you for your detailed review and questions. We’re happy to hear you appreciated the **“high accuracy”** resulting from a combination of a database with deep learning, and its **application to machine unlearning and model control at scale**.
>
> *How might the visual memory approach be adapted for tasks like object detection or segmentation?*
> Great question - we believe it is possible to extend to other tasks. In this paper, we represent each image with a single embedding, and this single embedding is obtained by pooling features from a spatial feature map at the top layers of an image encoder in models such as Dino or CLIP. While attention pooling, average pooling, or median pooling have been used in [previous work](https://openaccess.thecvf.com/content/ICCV2023/papers/Ranasinghe_Perceptual_Grouping_in_Contrastive_Vision-Language_Models_ICCV_2023_paper.pdf) to create a single embedding from these spatial feature maps, it is possible to pool these features into multiple embedding clusters instead of a single cluster using classical or learned clustering methods. As a proof of concept, we tested object segmentation based on a visual memory of DinoV2 features in response to your question. The result is visualized here: https://ibb.co/92dM0B2. This puts features from a *single* image into memory (a car in the example) based on 8 feature clusters and uses these to identify similar features in the second (test) image, thereby creating a segmentation mask. Such multi-vector representations of images (which could be expanded to image + text) are a simple and natural extension to our work in this paper, and will enable tasks such as object retrieval or detection from images containing multiple objects, or provide coarse patch-level semantic segmentation of images. Even finer-grained segmentation masks could be obtained with further training of the pooling methods. We hope this provides an intuition for how other tasks could be approached.
>
> *Reliance on a fixed pretrained embedding model may limit adaptability under significant distribution shifts.*
> Agreed, as we note in the discussion section, strong distribution shifts may require updating the embedding just as they would require re-training or fine-tuning a traditional deep learning model without memory. While our approach works well for typical distribution shifts (cf. Table 2), if the shift would become stronger, updating the embedding model may be advantageous. Possible options include incremental fine-tuning on new data and techniques from the adaptation literature (e.g., elastic weight consolidation).
>
> *Sensitivity of SoftmaxVoting to hyperparameters could impact performance stability.*
> You’re absolutely right, SoftmaxVoting - the current standard in the literature - is unstable and heavily depends on choosing a good setting for k (as we quantify in Tables 4–8 in the appendix) and also for the temperature (as quantified in Figure 9c). Therefore, e.g. the DinoV2 paper performs substantial hyperparameter sweeps to find hyperparameters that work well. For this reason, we propose to use RankVoting as a cheap, stable, and well-performing alternative that is much less sensitive to hyperparameters (as shown in Tables 4–8 and Figure 9a).
>
> *Retrieval storage & latency / Potential limitations in scalability for different embedding models with high memory footprints / Are there optimizations to manage storage costs of billion-scale datasets without affecting retrieval accuracy?*
> Please see https://openreview.net/forum?id=HoyKFRhwMS&noteId=YIxTk8VKZg showing that the memory footprint is only 1-3% of the cost of storing a dataset. Regarding optimizations, if storage is an issue then one could simply keep features while deleting the original dataset, saving 97-99% of storage cost. If one would like to optimize further, it may be worth training an embedding model with reduced feature dimensionality which would directly translate into storage savings.
>
> *Lack of performance benchmarking across diverse dataset distributions.*
> Could you elaborate what you’d like to see? We evaluate on ImageNet, iNaturalist, NINCO, InageNet-A, ImageNet-R, ImageNet-Sketch, ImageNet-V2, ImageNet-ReaL, i.e. eight different distributions currently.
>
> *It would be valuable to see an analysis of how the system performs when the memory contains corrupted samples*
> Thanks for the excellent suggestion. We conducted an analysis of RankVoting accuracy when the memory contains corrupted samples. The result is visualized here: https://ibb.co/X5d2PVg, showing that when 10%, 20%, 30%, …, all the way to 60% (!) of the labels in the memory database are corrupted (i.e., assigned to a random class), performance stays almost the same. We’ve added this plot to the manuscript in a dedicated appendix section (App. D, “Robustness towards label corruption”) and reference it from the main paper in Section 3.1. Thanks again for this suggestion which improves the paper.

---

> ### Author Response · Authors · 2024-11-20
> **Author response (2/2)**
>
> *Could you provide more details on how users might interact with or control the decision-making process?*
> Definitely. As a practical example, let’s say a visual memory based system is deployed in a clinical setting, and a radiologist uploads an X-ray scan. In contrast to a typical deep learning model, where the system would output a black-box prediction without explanation, a memory-based system could say “I think there are osseous fragments, because the scan looks very similar to the following 10 scans that I’ve seen previously. Here are these similar scans.” When looking at the evidence, the radiologist could then click an “exclude from prediction” button for any of the similar scans where their domain knowledge suggests that they shouldn’t be used in the decision-making process. This would then remove or downweigh the sample, just like the effect of hard vs. soft memory pruning described in Section 3.5 of our article, thereby altering the decision boundary.
>
> *Insufficient exploration of non-image applications (e.g., text-based memory) within the flexible memory paradigm.*
> A recent paper ([Scaling Retrieval-Based Language Models with a Trillion-Token Datastore](https://arxiv.org/pdf/2407.12854)) explores the exciting possibility of text-based memory. As implied by the title, the focus of our work is vision, but we now include a reference to this line of work.
>
> *What would be the computational cost for re-ranking with Gemini, particularly on constrained devices?*
> Great question. The computational cost for VLM-based re-ranking is a single VLM call. Naturally, different VLMs have a different computational footprint. Solutions for constrained devices include: 1.) using a VLM optimized for on-device inference, such as https://github.com/Meituan-AutoML/MobileVLM; 2.) hosting a more powerful VLM like Gemini on a server which is called from the device; in this case the bottleneck would no longer be the device itself but the communication bandwidth between device and server.
>
> *Could the proposed system be combined with incremental learning techniques to improve distribution shift resilience?*
> Thanks, that’s a great question. TL/DR: yes. There are at least three options: (1.) Since our approach uses standard model embeddings (e.g., DinoV2, CLIP) in a plug-and-play fashion, any incremental learning technique developed for standard models can simply be used to update the embedding for improved distribution shift resilience. (2.) That said, in contrast to existing models, adding new data to a visual memory does not come at the cost of poorer performance on previously seen data: since new knowledge can be added without fine-tuning, catastrophic forgetting of old data is not an issue since the data simply gets combined and the model maintains ready access to both new and old data - a clear advantage of a visual memory. (3.) Additionally, it may be possible to extend the memory in an ensemble fashion such that it can utilize information from a variety of trained models - including some updated through incremental learning techniques - to aggregate information for downstream tasks without requiring extra post-training.
>
> *How might RankVoting's performance vary with different embedding models?*
> Figures 2b (ImageNet) and 8b (iNaturalist) indicate that embedding model quality leads to a performance offset, but doesn’t change the quality / stability of scaling w.r.t. neighbors k. Furthermore, Tables 4–8 in the appendix show RankVoting performance across five different models. Does that address the question?
>
> *What measures could prevent sensitive or incorrect data from being memorized?*
> The same measures that can be applied to avoid training on sensitive or incorrect data - this is independent of the approach (classical training vs. building a memory). However, the clear advantage of using a visual memory is that it’s possible to easily remove data after the system has been trained / developed / deployed.
>
> *Could memory pruning methods be customized for specific tasks, improving model performance further?*
> Definitely, memory pruning can be customized. For instance, in medical imaging, one might prioritize pruning samples that are outdated, while in autonomous driving, one might prioritize pruning samples that lead to misclassification of pedestrians. Tailoring pruning strategies to specific tasks could further improve model performance and address domain-specific challenges.
>
> *How does the model handle cases where similar but incorrect neighbors dominate?*
> If the model is queried with a cat image, and 8 out of the nearest 10 neighbors are very similar cat images, then the model would respond with the label of those 8 images. If this label is incorrect, then that suggests either a dataset (=label) problem or an embedding model problem, both of which are independent of our approach - our visual memory benefits from better data and better embeddings.
>
> Thanks again for the questions & suggestions!

---

> > ### Comment · Area_Chair_DEmo · 2024-11-25
> >
> > Dear reviewer,
> >
> > The authors have provided another round of responses. Could you kindly review them and provide your feedback?
> > Thanks

---

> > > ### Comment · Reviewer_S9w1 · 2024-11-26
> > > **After Rebuttal**
> > >
> > > Thanks to the authors for the detailed responses. The explanations from Rebuttal have partly addressed my concerns.
> > >
> > > I agree with the argument regarding other reviewers, and thus I have to keep my **initial** rating.

---

### Official Review · Reviewer_pVvh · 2024-11-02

**Soundness:** 4
**Presentation:** 4
**Contribution:** 3
**Rating:** 8
**Confidence:** 4

**Summary:**

The authors address the memory-based image classification task by utilizing large, unsupervised pretrained visual models to extract image descriptors. They perform classification based on nearest neighbor retrieval in the embedding space of these models. A key contribution is their novel aggregation approach called _RankVoting_, which outperforms state-of-the-art methods. Unlike previous methods, _RankVoting_ improves performance as the number of considered neighbors increases. Through various experiments, the authors demonstrate the advantages of visual memory-based classification over traditional model-based classification, highlighting benefits such as flexibility, scalability, and interpretability.

**Strengths:**

* __Presentation.__ The paper is written in a highly engaging and accessible manner, capturing the reader's interest from the outset. It is well-positioned within the context of related works, providing a clear understanding of how it contributes to the field. The novelty of the research is evident, and the authors effectively communicate the motivation behind each experiment.

* __Experimental design.__ The authors have conducted a comprehensive set of experiments that excellently demonstrate the advantages of memory-based image classification over traditional model-based approaches. They show how their method easily scales to accommodate new data, illustrating its effectiveness in scenarios requiring continual learning. Furthermore, they highlight the flexibility of their approach in machine unlearning and memory pruning, showcasing how it can efficiently remove or adjust the influence of specific data points. The interpretability of their method is also emphasized, providing clear insights into the decision-making process of the model.

* __Novelty.__ The main highlight of the paper is the introduction of the novel neighbor aggregation method called RankVoting. This method is clearly and thoroughly described, with an intuitive rationale that makes its underlying principles easy to grasp. The authors present solid empirical evidence of its advantages over state-of-the-art distance-based softmax voting methods, demonstrating superior performance, especially as the number of considered neighbors increases. This contribution significantly advances the field by providing a more effective way to aggregate information from nearest neighbors in neighbor-based classification.

**Weaknesses:**

While some may argue that the methodological novelty of the paper is modest—introducing only a single, simple formula to slightly improve kNN classification—I believe that this advancement is substantial in its own right. The elegance and simplicity of the _RankVoting_ method support its effectiveness, providing a meaningful enhancement to nearest neighbor classification. Moreover, the authors contribute significantly to the broader discussion of neighbor-based versus model-based image classification. They present strong, well-articulated arguments that underscore the advantages of visual memory-based approaches. The paper not only provides a practical improvement but also stimulates important discourse that could shape future research directions.

**Questions:**

In Table 2, for the ImageNet-A dataset, the performance achieved through linear probing surpasses even that of the JFT memory combined with Gemini reranking. This is not the case for the other datasets. Initially, I considered that this anomaly might indicate some form of overfitting. However, later in the paper you mention potential issues with the ground-truth labels in the ImageNet-A dataset, which could affect the trend.

Did you observe similar issues with any of the other datasets used in your experiments? Moreover, do you think it is possible to quantify this problem—for instance, by assessing how reassigning or correcting problematic labels influences the performance?

---

> ### Author Response · Authors · 2024-11-20
> **Author response**
>
> Dear Reviewer pVvh,
>
> Thank you for your review, we are happy to hear that you found the paper presenting **“strong, well-articulated arguments”** for visual memory, **“stimulating important discourse that could shape future research directions”**, with **“excellent soundness & presentation”**.
>
> Regarding your question about potential issues with the ground-truth labels in the ImageNet-A dataset, and whether it may be possible to quantify this problem: That’s a great suggestion. To address this, we performed a human experiment on a randomly selected subset of ImageNet-A (N=100) where the dataset label and the prediction from DinoV2 ViT-L14 with JFT memory disagree. We presented the image alongside the original ImageNet-A label and our model-predicted label to three human observers, asking them to identify which of the labels best describes the image (of course, without telling them which of the labels is the dataset label). The result was that in 39.3% (!) of cases (std: +/- 1.25%), the DinoV2 label was assessed as being better/more suitable than the original dataset label - i.e., roughly 2 out of 5 model “errors” are in fact dataset label errors, quantifying the ImageNet-A label quality issue we alluded to in Figure 6. This percentage can be used to estimate how correcting problematic labels influences performance. Instead of the original model’s 61.1% accuracy on ImageNet-A, due to label errors the ‘corrected’ accuracy is instead 76.4% (a delta of +15.3% in absolute terms or +25.0% in relative terms). We believe that the issue is indeed limited to ImageNet-A label quality. Thanks again for the suggestion; we have added a discussion on this important issue to a dedicated appendix section (App. M - "ImageNet-A error analysis") and link the human experiment and result from Section 3.7 of the main paper.

---

> > ### Comment · Reviewer_pVvh · 2024-11-24
> >
> > I would like to thank the authors for the additional experiments and discussions provided. The extra analyses and clarifications have addressed my concerns effectively. I will maintain my current rating for this submission.

---

> > > ### Author Response · Authors · 2024-11-25
> > > **Thanks!**
> > >
> > > Thanks for letting us know, we're glad to hear that your concerns were addressed.

---

### Author Response · Authors · 2024-11-20
**Latency and storage**

Several reviewers asked about the latency and storage requirements of our approach - excellent questions that we are happy to address since those are important considerations that improve our manuscript.

**TL/DR: Storing features requires only about 1-3% of the space of storing the dataset itself; in terms of latency ScaNN can handle 500-600 queries per second for a million-scale memory.**

The detailed answer - now added as a dedicated section in the Appendix (App. J, "Storage and latency") and referenced in Section 2.2 of the main paper under “fast and scalable nearest neighbor search”:

**Latency.** Nearest neighbor retrieval, fortunately, does not need to reinvent the wheel but can, instead, build on top of highly optimized workloads and libraries such as the ScaNN library (Guo et al. 2020). The [ScaNN github README](https://github.com/google-research/google-research/tree/master/scann) shows a latency comparison - see here for a latency comparison: https://ibb.co/JnT8N2n; with the requirement of perfect recall a million-size memory can handle roughly 500-600 queries per second. It may be worth mentioning that searching a large database can be done on CPUs and can be heavily parallelized. We are working towards adding more scaling latency analyses and will update the appendix section once we have them.

**Storage.** In addition to latency, storage is another very practical consideration: How much does it take to store features for a large database? To put things into perspective, the ImageNet training dataset requires 154.6 GB of storage, and the ImageNet validation dataset requires 6.0 GB of storage. In comparison, as shown in this table here: https://ibb.co/zxbjHWm (anonymous link), storing DinoV2 or CLIP features for the entire ImageNet training dataset only requires between 1.9 and 4.9 GB of storage space. Thus compared to storing the training dataset, the model features account for only 1-3% of this size. This means that after constructing the memory, one may decide to keep the dataset which adds 1-3% of storage, or one may decide to delete the dataset only keeping the features which saves 97-99% of storage (compared to the dataset storage requirement). The ratio of features requiring 1-3% of the dataset size doesn’t change with dataset scale since it only depends on the embedding model, thus this ratio would hold for very small datasets just as it would for a billion-scale dataset.

---

### Author Response · Authors · 2024-11-20
**Author's summary of rebuttal discussion**

We would like to thank all reviewers for their valuable feedback and we very much appreciate their assessment of our work as
- **“excellently demonstrating the advantages of memory-based image classification”** & **“stimulating important discourse that could shape future research directions”** (pVvh),
- **“addressing practical needs”** and **“achieving high accuracy and flexibility”** (S9w1),
- **“well-written”** with **“particularly interesting OOD experiments”** (AsSs)
- **“strong and interesting Gemini re-ranking results”** and a **“promising strategy for visual recognition”** (X1Pn).

We received excellent suggestions for improvement which we feel will substantially strengthen the paper; here is a summary of main suggestions/concerns and how we addressed them:
- Quantifying model vs. label errors on ImageNet-A through human experiment (pVvh)
- Proof of concept for extending approach to object segmentation (S9w1)
- Adding latency & storage statistics (S9w1, AsSs, X1Pn)
- Analyzing performance under label corruption showing very strong resilience (S9w1)
- Clarifying linear probe setup & main contribution (AsSs)
- Analyzing effect of class imbalance on accuracy (AsSs)
- Clarifying that accuracy increases with memory scale (X1Pn)

Thanks again to all reviewers for their time and feedback!

---

### Meta-Review · Area_Chair_DEmo · 2024-12-15

**Metareview:**

This work introduces a flexible and interpretable visual memory system that combines deep neural network embeddings with database-like capabilities, enabling scalable data addition, unlearning, and intervention for controlled decision-making.

The paper is generally well-written, presenting an intriguing idea of utilizing external memory to enable flexible and adaptive modifications to AI models. This approach holds significant potential for various applications, including data addition, unlearning, and intervening in decision-making.

However, the paper received mixed review scores: 8, 6, 6, and 3.

After reviewing all the feedback, the AC identified key unresolved issues raised by Reviewer AsSs, which ultimately led to the paper's rejection. Despite the rebuttal, Reviewer AsSs maintained two primary concerns:

[unclear novelty] Many works have presented similar claims or methods (for example, the "simple scales well" claim => https://arxiv.org/pdf/1702.08734 did billion-scale k-nearest-neighbour search with image features in 2017). It is unclear what is the technical novelty or contribution of this paper.

[professional writing] The claims in the paper are not written in a scientific manner (claims like the algorithm being "Lightning fast." (line 356-357) are not statements with any scientific meaning).

**Additional Comments On Reviewer Discussion:**

The paper received mixed review scores: 8, 6, 6, and 3.

After reviewing all the feedback, the AC identified key unresolved issues raised by Reviewer AsSs, which ultimately led to the paper's rejection. Despite the rebuttal, Reviewer AsSs maintained two primary concerns:

[unclear novelty] Many works have presented similar claims or methods (for example, the "simple scales well" claim => https://arxiv.org/pdf/1702.08734 did billion-scale k-nearest-neighbour search with image features in 2017). It is unclear what is the technical novelty or contribution of this paper.

[professional writing] The claims in the paper are not written in a scientific manner (claims like the algorithm being "Lightning fast." (line 356-357) are not statements with any scientific meaning).

---

### Decision · Program_Chairs · 2025-01-22

Reject